# Onset of persistent surface ocean oxygenation during the Great Oxidation Event

Andy W. Heard [1] ✉, Chadlin M. Ostrander [2], Yunchao Shu [3], Andrey Bekker [4,5], Simon W. Poulton [6] & Sune G. Nielsen[1,7]

Free oxygen ($O_2$) first began accumulating in Earth's atmosphere shortly after the Archean-Proterozoic transition during the 'Great Oxidation Event' (GOE). The nature of surface ocean oxygenation at this time is, however, poorly quantified, limiting our understanding of planetary oxygenation thresholds. Geochemical records of global ocean redox may potentially shed light on this critical interval. Here, we show that vanadium (V) isotope ratios in 2.32-2.26-billion-year-old (Ga) shales from the Transvaal Supergroup, South Africa, capture a unidirectional transition in global ocean redox conditions shortly above the stratigraphic level marking the canonical rise of atmospheric $O_2$. Around 2.32 Ga, sedimentary sinks were dominated by anoxic environments that drove extensive seawater V drawdown. A positive shift in seawater V isotopic composition in the overlying stratigraphy indicates global expansion of marine settings with $\geq 10\ \mu M$ dissolved $O_2$ in bottom water, likely restricted to shallow-water environments and attributable to widespread equilibration with an oxygenated atmosphere.

The rise of atmospheric $O_2$ partial pressure ($pO_2$) during the GOE, ca. 2.43-2.22 billion years ago[1–3], marks the permanent oxygenation of Earth's surface. Despite intense study of this time interval, there is little consensus on the tempo and amplitude of $O_2$ accumulation in the atmosphere[4], and even less consensus on its accumulation in the ocean[5]. Marine oxygenation in response to the GOE fundamentally changed the trajectory of biological innovation on Earth, ultimately laying the groundwork for complex multicellular life, and constituted a critical step in defining the ultimate nature of Earth's habitability[6,7].

The GOE is marked most clearly by loss of sulfur isotope mass-independent fractionation (S-MIF) signatures, generated by photochemical reactions in an oxygen-free atmosphere, from the sedimentary record[8]. After the GOE, there was a sufficient $pO_2$, above a threshold of $>10^{-6}$ of the present atmospheric level (PAL), to prevent preservation of S-MIF signatures[9]. The disappearance of S-MIF was initially thought to occur as a unidirectional 1–10 million year transition[2,3], approximately corresponding to the boundary separating the Rooihoogte and Timeball Hill formations of the Transvaal Supergroup, South Africa, dated to $2.316 \pm 0.007$ Ga[10] (Fig. 1A). Subsequent work has provided possible evidence for older and younger Paleoproterozoic S-MIF disappearances[1,11,12]. The significance of younger S-MIF is still debated (see Supplementary Information for points of relevance to this study), but multiple possible fluctuations across the GOE interval suggest that atmospheric $pO_2$ may have oscillated across the $10^{-6}$ PAL threshold during a transition lasting from ca. 2.43 to 2.22 Ga[1,12–14].

The ca. 2.43–2.22 Ga time interval encapsulates a knowledge gap in the marine response to the GOE. A widely recognized signal for marine biogeochemical overhaul is not seen until the ca. 2.22–2.06 Ga Lomagundi carbon isotope excursion (LCIE), the largest positive

[1]Department of Geology & Geophysics, Woods Hole Oceanographic Institution, Woods Hole, MA, USA. [2]Department of Geology & Geophysics, University of Utah, Salt Lake City, UT, USA. [3]Laoshan Laboratory, Qingdao, China. [4]Department of Earth and Planetary Sciences, University of California, Riverside, CA, USA. [5]Department of Geology, University of Johannesburg, Auckland Park, South Africa. [6]School of Earth and Environment, University of Leeds, Leeds, UK. [7]CRPG, CNRS, Université de Lorraine, Vandoeuvre lès Nancy, France. ✉e-mail: andrew.heard@whoi.edu

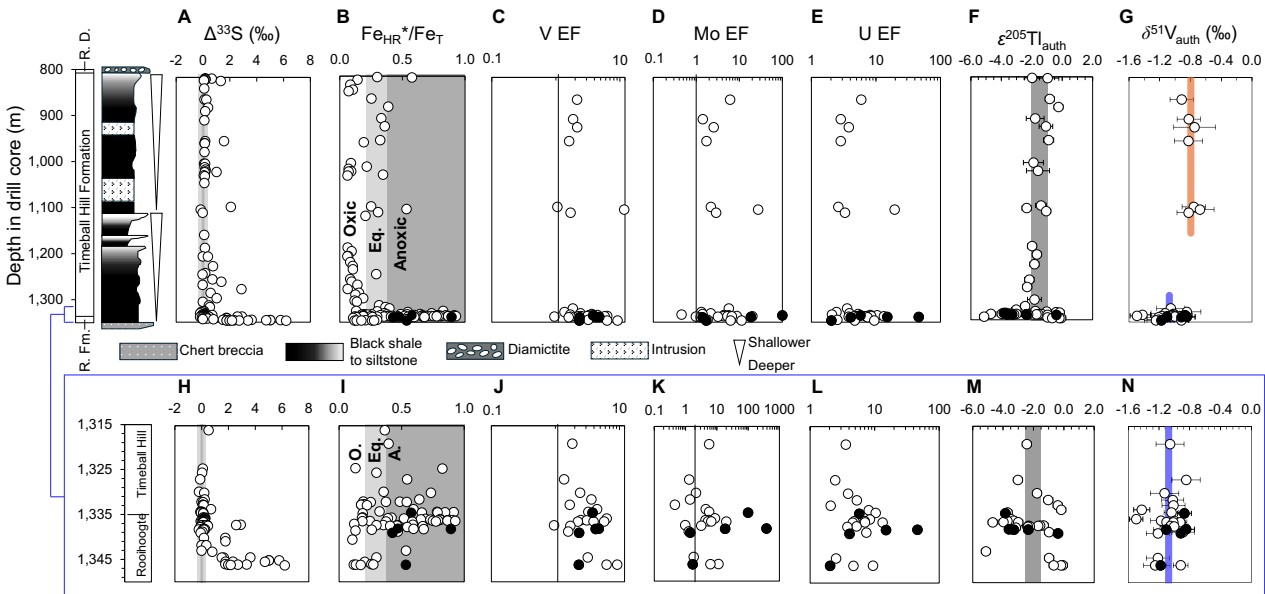

**Fig. 1 | Geochemistry of the EBA-2 drill core, South Africa, deposited during the GOE. A** Fluctuating sulfur isotope mass-independent fractionation (S-MIF) ($\Delta^{33}S$) signature tracks possible oscillations in atmospheric $O_2$ levels, potentially up until ca. 2.22 Ga[2,3,12]. **B** Elevated $Fe_{HR}*/Fe_T$ values, which distinguish oxic ('O.'; <0.22), equivocal ('Eq.'; >0.22 to <0.38) or anoxic ('A.'; >0.38) water-column redox conditions[12], in addition to (**C**) Vanadium, (**D**) Molybdenum, and (**E**) Uranium enrichment factors (EF), all designate that a subset of shales was deposited under anoxic (ferruginous or euxinic) conditions. **F** Authigenic Tl isotope ratios ($\varepsilon^{205}Tl_{auth}$) provide information about ocean redox conditions, specifically Mn-oxide burial[16]. The gray-shaded region shows the estimated average for upper continental crust[18]. **G** Authigenic V isotope ratios ($\delta^{51}V_{auth}$) that can be used to infer paleo-$\delta^{51}V_{sw}$ values using known isotopic differences between seawater and sediments in anoxic environments. **H–N** show the same data as for (**A–G**), with the narrow stratigraphic range at the base of the section from 1315 to 1350 drill core depth expanded for clarity. Black-filled datapoints highlight samples analyzed in this study that have $Fe_{py}/Fe_{HR}*$ ratios >0.6, indicating possibly euxinic conditions. Error bars for $\delta^{51}V_{auth}$ are 2 SD of reproducibility on either the individual sample or the BDH chemicals V solution standard, whichever is larger. The orange and blue vertical lines show the average $\delta^{51}V_{auth}$ values for the upper Timeball Hill Formation (<1300 m drill core depth) and the upper Rooihoogte and lower Timeball Hill formations (>1300 m drill core depth), respectively. R. Fm. – Rooihoogte Formation and R. D. – Rietfontein Diamictite.

carbonate carbon isotope excursion in Earth history, which has been linked to unprecedented organic carbon burial that drove ocean oxidation[15]. Evidence of coupled atmospheric and marine oxygenation was reported at the ca. 2.32 Ga Rooihoogte-Timeball Hill formation boundary (Fig. 1), with thallium (Tl) isotopic data indicating widespread burial of Mn oxides, requiring oxygenated bottom waters on shallow marine shelves at almost the same stratigraphic level where persistent, large S-MIF signals disappear[16].

However, evidence for Mn oxide burial provides only one, qualitative index for rising $O_2$, so to provide further texture to our understanding of ocean oxygenation across the GOE, we measured sedimentary V isotope ratios (reported as $\delta^{51}V = ({}^{51/50}V_{sample}/{}^{51/50}V_{AA\,Specpure} - 1) \times 1000$) in the same Rooihoogte and Timeball Hill shale samples previously analyzed for Tl isotope values[16]. Vanadium isotopes track the global marine redox state as Tl isotope values do, but the oxidized sink for V in the oceans records a threshold dissolved $O_2$ level[17] (>10 µM; outlined in detail below), rather than the specific burial of Mn oxides[18]. As such, combined V and Tl isotopic data can provide more nuance to reconstructions of global ocean oxygenation events and their impacts on multiple redox-sensitive element cycles[19]. In this study, we targeted organic-rich shales and analyzed $\delta^{51}V$ in the authigenic V fraction ($\delta^{51}V_{auth}$). This fraction represents the V scavenged from Paleoproterozoic seawater by sinking organic matter, the isotopic composition of which allows reconstruction of relative changes in the global ocean redox state[20].

Vanadium isotope geochemistry provides information on the global ocean redox state because 1) V is redox sensitive, behaving differently and taking on different isotopic compositions in different redox environments (Fig. 2); and 2) it has a long (ca. 90 kyr) residence time relative to modern and ancient ocean mixing timescales on the

order of 1 kyr[21], such that the seawater dissolved V reservoir and its isotopic signature should be globally well-mixed in the open ocean and unrestricted basins[20]. This global homogeneity is also expected to hold in the Paleoproterozoic, because V concentrations in black shales from this time period suggest a comparable order-of-magnitude size of the marine dissolved V pool to the modern[22,23], and thus there is no obvious reason why the oceanic V residence time would be orders of magnitude lower[19]. Because of this, reconstruction of marine V isotope mass balance from sedimentary archives can shed light on the ancient ocean redox state.

Dissolved V is deposited in sediments with isotopic differences relative to open-ocean seawater ($\Delta^{51}V = \delta^{51}V_{sed} - \delta^{51}V_{sw}$) that are controlled by local redox conditions[17,20] (Fig. 2). These differences commonly result in the enrichment of sediments in isotopically light V relative to seawater, and they decrease in magnitude in more reducing environments[20]. On continental margins and abyssal plains where bottom-water dissolved $O_2$ concentrations exceed 10 µM, hydrogenous Fe-Mn oxyhydroxides and pelagic clay sediments exhibit an isotopic difference $\Delta^{51}V_{O2 > 10\,\mu M}$ from seawater of ca. –1.1 ± 0.1‰ that can be explained by isotopic equilibrium between vanadate [V(V)] dissolved in seawater and adsorbed onto a range of Fe−Mn oxyhydroxide surfaces[20,24]. Under reducing conditions, vanadate is reduced to vanadyl [V(IV)], which has a strong affinity for organic carbon particles. In reducing open-ocean settings, sediments commonly have a $\Delta^{51}V$ of ca. –0.7‰[17]. The driver of this isotope fractionation is almost certainly ${}^{50}V$-rich vanadyl incorporation into sinking organic matter[25–27], although whether this represents an equilibrium or kinetic fractionation is yet to be determined. In sediments from reducing, restricted, and typically euxinic (anoxic and $H_2S$-rich) basins like the Cariaco Trench, $\Delta^{51}V$ is ca. –0.4‰ relative to seawater[17,28]. This smaller

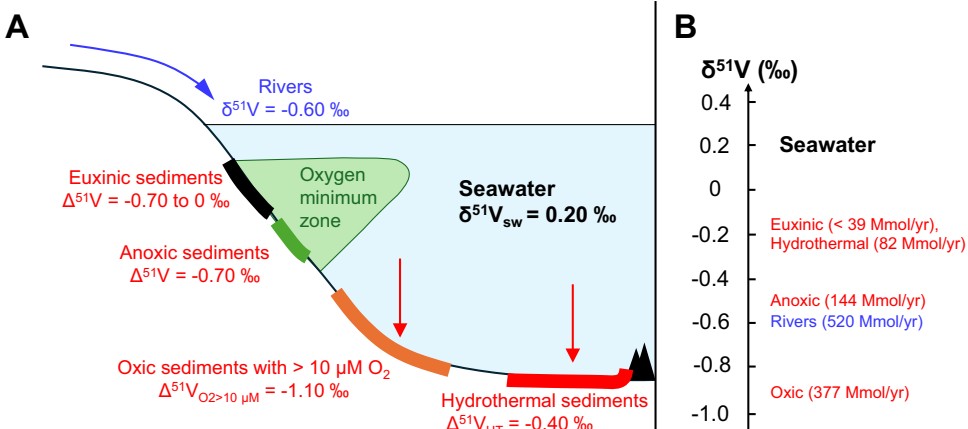

**Fig. 2 | Modern V isotope mass balance in the oceans. A** Schematic diagram. Rivers are the dominant input to the ocean (blue arrow). Red arrows show sedimentary removal pathways from seawater, and the associated isotopic difference ($\Delta^{51}V$) of each sink relative to seawater. The modern ocean residence time for V is ca. 90,000 yr. **B** Vanadium fluxes in Mmol/yr and isotopic compositions in $\delta^{51}V$ (vertical position) corresponding to sources and sinks represented in panel A. The $\delta^{51}V$ of modern seawater is highly positive today relative to riverine inputs because of the dominant sedimentary V removal flux to oxic sediments with strongly negative $\delta^{51}V$. Figure featuring compiled data from ref. 20.

difference is unrelated to further reduction of V, which is kinetically inhibited[29], or V-sulfide formation, which requires extreme $H_2S$ concentrations[30]. Rather, the same instantaneous isotopic difference between seawater and organic particles of −0.7‰ can explain the offset of Cariaco sediments from global seawater by considering the 65% drawdown of seawater V in the basin in the context of a Rayleigh distillation model[17]. Full drawdown could quantitatively sequester seawater V, so $\Delta^{51}V$ values between reducing sediment and global seawater over the course of Earth history likely ranged from −0.7 to 0.0‰[20]. Lastly, the co-precipitation of vanadate with $Fe^{3+}$-oxyhydroxides formed during hydrothermal venting is associated with an isotopic difference $\Delta^{51}V_{HT}$ of ca. −0.4‰[31], but $O_2$ at mid-ocean ridge depths is not expected to have been present during the GOE to support rapid hydrothermal Fe oxidation and V coprecipitation[32,33]. Hydrothermal fluid V input to the oceans is also expected to have a negligible effect on the seawater isotope mass balance[34].

Most V enters the ocean via rivers, with $\delta^{51}V = -0.6 \pm 0.1$‰, matching that of upper continental crust (UCC) igneous rocks[20,35,36] (Fig. 2). The early Paleoproterozoic UCC, and thus likely also the riverine V input, was isotopically lighter by at least 0.1‰ due to a difference in the magmatic character of primary igneous rocks[37]. Modern open marine $\delta^{51}V_{sw}$ is globally homogeneous at around $0.20 \pm 0.07$‰[36,38,39], and is significantly more positive than the riverine input because the dominant sink of V is oxidized sediments with their large negative $\Delta^{51}V_{O2>10\mu M}$[20,38] (Fig. 2). In a more reducing global ocean, $\delta^{51}V_{sw}$ should decrease as the V output flux shifts towards reduced sinks with comparatively smaller $\Delta^{51}V$ values[19,20,40,41]. Inferring changes in $\delta^{51}V_{sw}$ on ancient Earth is hindered by the lack of unfractionated sedimentary archives, so targeting sediments with well-characterized depositional redox conditions is thus necessary to correct for offsets between sediment and coeval $\delta^{51}V_{sw}$.

We analyzed samples from 850 to 1,346 m depth in well-preserved drill core EBA-2 drilled near Carltonville, South Africa (Fig. 1; see "Methods" for further information). These samples document the Rooihoogte and Timeball Hill formations, which were deposited in a pro-deltaic environment interpreted to have been connected to the open ocean[12,42]. The canonical disappearance of S-MIF occurs in the upper Rooihoogte Formation at -1,340 m drill core depth[2], with a depositional age of $2.316 \pm 0.007$ Ga[10]. Further age constraints come from U−Pb dating of two tuff beds in the nearby drill core EBA-1, which gave ages of $2.256 \pm 0.006$ Ga and $2.266 \pm 0.004$ Ga for the upper Timeball Hill Formation, representing the top of our studied section[43]. Geochemical indicators from the EBA-2 drill core show that local redox

conditions fluctuated between oxic and anoxic throughout deposition of the Rooihoogte and Timeball Hill formations[12,16] (Fig. 1).

Almost all samples we targeted have highly reactive Fe to total Fe ratios higher than those expected for oxic conditions ($Fe_{HR}^*/Fe_T > 0.22$, where $Fe_{HR}^*$ includes a correction for highly reactive Fe incorporated into clay minerals during diagenesis; Fig. 1B), and most have $Fe_{HR}^*/Fe_T > 0.38$, which is above the calibrated threshold for anoxic deposition[12,44]. A handful of samples in the Rooihoogte Formation have pyrite to total reactive iron ($Fe_{py}/Fe_{HR}^*$) values that place them in the possibly euxinic ($Fe_{py}/Fe_{HR}^* = 0.6–0.8$) or euxinic ($Fe_{py}/Fe_{HR}^* > 0.8$) fields[45]. Additionally, bulk samples generally feature elevated enrichment factors (X EF) relative to the UCC (X EF = $(X/Al)_{sample}/(X/Al)_{UCC}$) for the redox-sensitive elements V, Mo and U[16], which requires reducing (and in the case of Mo, euxinic) conditions[46–49]. Two samples in the lower part of the section with $Fe_{HR}^*/Fe_T < 0.22$, which potentially indicates deposition under oxic conditions, have V, Mo and U EFs much higher than 1, suggesting deposition under reducing conditions, where $Fe^{2+}$ was remobilized back to the water column from anoxic non-sulfidic porewaters at the sediment-water interface[45].

Despite capturing an important interval of Earth history in the immediate aftermath of the first S-MIF disappearance, we did not analyze authigenic V isotopes in the stratigraphic interval between -1100 and 1300 m depth. All available samples from this interval were likely deposited under oxic conditions based on $Fe_{HR}^*/Fe_T$ ratios that are lower than 0.22 (Fig. 1B). This sedimentary redox condition requires different leaching procedures due to a different dominant host phase for authigenic V (vanadate adsorbed to Fe oxyhydroxides), and this leaching procedure has yet to be applied to or calibrated for ancient sedimentary rocks, having only been tested in modern marine sediments[17].

## Results and discussion
The $\delta^{51}V_{auth}$ values range from −1.50 to −0.68‰ in the Rooihoogte and Timeball Hill formations (Fig. 1G, Supplementary Data S1). Much of this variability comes from a few outliers low in the stratigraphy; the drivers of which are discussed below and in Fig. S1. The samples define two populations based on their stratigraphic position. In the samples deposited deeper than 1,300 m drill core depth (hereafter referred to as the 'lower section'), the average value of $\delta^{51}V_{auth}$ is −1.07 ± 0.07‰ (2SE). In the samples from above 1100 m drill core depth (hereafter referred to as the 'upper section'), the average value of $\delta^{51}V_{auth}$ is −0.80 ± 0.05‰ (2SE). There is therefore an increase in the average $\delta^{51}V_{auth}$ value of 0.27 ± 0.12 ‰ between the lower and upper sections.

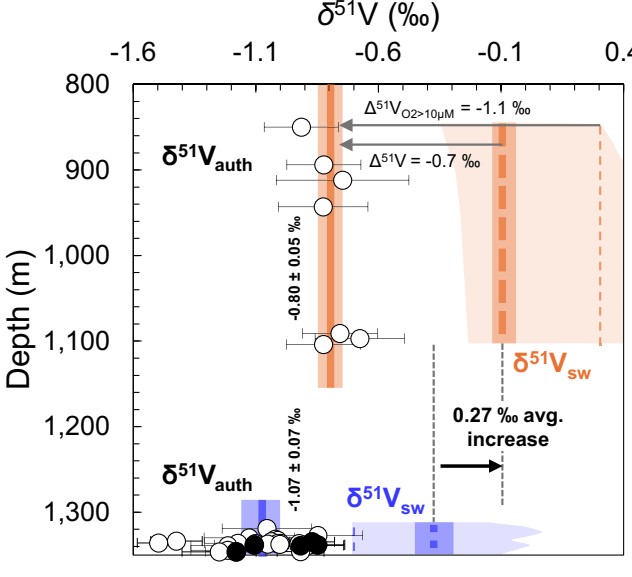

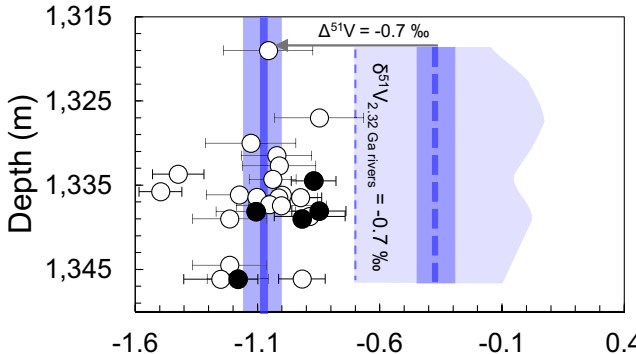

**Fig. 3 | Estimated seawater vanadium isotopic compositions for the Rooihoogte and Timeball Hill formation shales.** Authigenic $\delta^{51}V$ data are shown as in Fig. 1. Orange and blue solid lines show the average $\delta^{51}V_{auth}$ values for the upper Timeball Hill Formation (<1300 m drill core depth) and the upper Rooihoogte and lower Timeball Hill formations (>1300 m drill core depth). Darker orange- and blue-shaded boxes surrounding thick dashed lines indicate estimated allowable ranges of average $\delta^{51}V_{sw}$ in each section, within local redox constraints and with a minimum value imposed by a syn-Great Oxidation Event (GOE) riverine input of −0.7‰ based on the average composition of the UCC at this time[37]. Lighter shaded regions show the range of $\delta^{51}V_{sw}$ permitted by the analytical uncertainty of individual sample analyses. For the upper section, $\Delta^{51}V$ values between −1.1 and −0.7‰ are permitted by local redox indicators consistent with O₂ around a 10 μM threshold value[17]. For the lower section, persistently anoxic (and sometimes euxinic) conditions define a range of $\Delta^{51}V$ extending no lower than −0.7‰[17]. An increase in average $\delta^{51}V_{sw}$ of 0.27 ± 0.12‰ would be defined if the same local $\Delta^{51}V$ were applied to the upper and lower sections. Because the upper section was deposited under locally more oxidizing conditions (that permit a larger $\Delta^{51}V$), this inferred increase in $\delta^{51}V_{sw}$ can be robustly considered as a minimum. Error bars for $\delta^{51}V_{auth}$ are 2 SD of reproducibility on either the individual sample or the BDH chemicals V solution standard, whichever is larger.

## Reconstructing paleoseawater $\delta^{51}V$ evolution during the GOE

As discussed above, the targeted samples were deposited under reducing conditions. These local conditions make it appropriate to first consider an effective $\Delta^{51}V$ between paleoseawater and sediments in a spectrum of values between −0.7 and 0.0 ‰, depending on the extent of local V drawdown (Fig. 3)[20]. The $\delta^{51}V_{auth}$ values show no correlation with $Fe_{HR}^*/Fe_T$, $Fe_{py}/Fe_{HR}^*$, or total organic carbon (TOC) (Fig. S1). A possible negative co-variation of $\delta^{51}V_{auth}$ with V EF, and apparent positive correlation of TOC with V EF, occur in the lower

section, which may indicate water column V depletion that plausibly could be driven by organic matter (Fig. S1). However, this relationship is weak relative to those commonly seen in younger Precambrian shales[40], making it difficult to select a specific value for $\Delta^{51}V$ (within the range of −0.7 to 0.0‰) to reconstruct $\delta^{51}V_{sw}$, or to apply a variable sample-by-sample correction. Related to this, much of the scatter in $\delta^{51}V_{auth}$ in the lower section may reflect variability in the local $\Delta^{51}V$ expressed during V drawdown to sediments under reducing conditions, relative to a potentially less variable $\delta^{51}V_{sw}$ at the time of deposition. The approximate minimum reconstructed value of $\delta^{51}V_{sw}$ allowable by mass balance should be defined by the syn-GOE riverine value, as no known sinks decrease $\delta^{51}V_{sw}$ relative to inputs. Based on the -0.1‰ lighter composition of the UCC at this time, we assume the riverine value and thus the minimum allowable paleo-$\delta^{51}V_{sw}$ was around −0.7‰[36,37] (Fig. 3). Meanwhile, the maximum paleo-$\delta^{51}V_{sw}$ during deposition of the lower section is defined by a 0.7‰ offset to the $\delta^{51}V_{auth}$ data, giving a maximum value of -0.37‰ on average, with the range allowed within analytical error on individual data points extending to higher values (Fig. 3).

In the upper section, low V EF values appear to be related to low TOC contents (Fig. S1), rather than water column V drawdown, with Fe speciation and redox-sensitive element data generally indicating conditions that were reducing, but not as reducing as in the lower section. Most upper section samples feature 'equivocal' Fe speciation values $0.22 < Fe_{HR}^*/Fe_T < 0.38$ (Fig. 1B), similar to those seen on the Peruvian margin[17] in environments with bottom seawater O₂ content straddling 10 μM. Therefore, we can consider the effect of applying a maximum magnitude of correction equal to $\Delta^{51}V_{O2>10\,\mu M} = -1.1‰$ to these samples (Fig. 3). We assume the smallest possible value for $\Delta^{51}V$ to be −0.7‰ for the upper section, as seawater V drawdown was likely not extensive under those conditions. When applying this range of fractionation factors, the average paleo-$\delta^{51}V_{sw}$ for the upper section would be −0.1 to +0.3‰ (Fig. 3).

Due to the large range of absolute $\delta^{51}V_{sw}$ values that can be reconstructed for the Rooihoogte and Timeball Hill formations (Fig. 3), below we focus on the implications of stratigraphic changes rather than these absolute values. A uniform application of the same local $\Delta^{51}V$ correction (e.g., the reducing open-ocean value $\Delta^{51}V = -0.7‰$) results in an average increase in the $\delta^{51}V_{sw}$ value of 0.27 ± 0.12‰ going from the lower to upper section. However, as shown in Fig. 3, a larger positive shift in $\delta^{51}V_{sw}$ from the lower to upper section is conceivable. While local redox conditions in the sections are variable, the lower section generally indicates more reducing conditions than the upper section, with higher $Fe_{HR}^*/Fe_T$ ratios and V and U EFs, as well as higher $Fe_{py}/Fe_{HR}^*$ ratios indicative of potentially euxinic conditions. If a variable, local-redox-dependent $\Delta^{51}V$ correction were to be applied to the target shales, larger corrections would be required for the less reducing upper section. This would result in a substantially larger up-section increase in the inferred $\delta^{51}V_{sw}$. Furthermore, despite deposition in an environment that was well-connected to the open ocean, the pro-delta setting may conceivably have allowed some degree of mixing between global seawater and UCC-like river water inputs, which could bias reconstructed $\delta^{51}V_{sw}$ towards slightly more negative values than the real seawater value[36]. This bias, if present, would have more significantly impacted the samples from the top of the section that were deposited closest to shore under the shallowest paleodepths (Fig. 1), so any correction to account for this would again only increase the magnitude of the positive shift in reconstructed $\delta^{51}V_{sw}$ up-section.

## Ocean redox response to rising $pO_2$

What are the implications of a ≥0.27‰ positive shift in $\delta^{51}V_{sw}$ in the stratigraphy above the canonical disappearance of S-MIF? The largest isotopic lever that operates in marine V cycling is the oxidized vanadate sink, with $\Delta^{51}V_{O2>10\,\mu M} = -1.1‰$. Under increasingly reducing conditions, which favor vanadyl drawdown by sinking organic matter,

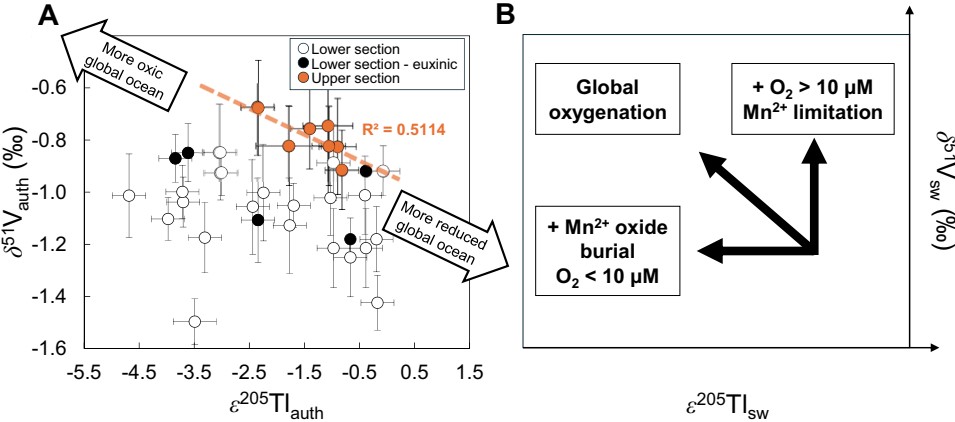

**Fig. 4 | Cross-plot of V and Tl isotopic data for the Rooihoogte and Timeball Hill formation shales and expected global redox trends. A** Cross-plot of $\delta^{51}V_{auth}$ and $\varepsilon^{205}Tl_{auth}$ data for shales from the lower section (white symbols, euxinic/possibly euxinic samples in black symbols) and upper section (orange) symbols. The upper section features a negative correlation, with a small overall range in $\varepsilon^{205}Tl_{auth}$, while the lower section features greater scatter that may reflect variations in the value of $\Delta^{51}V$ during deposition. **B** Expected qualitative trends for $\delta^{51}V_{sw}$ and $\varepsilon^{205}Tl_{sw}$ during different modes of ocean oxygenation. Global oxygenation that sees a broad expansion of both environments with $O_2 > 10\,\mu M$ and Mn-oxide burial should result in increased $\delta^{51}V_{sw}$ and decreased $\varepsilon^{205}Tl_{sw}$. Negative trends in (**A**) suggest some degree of global coupling of the V and Tl cycles. Significant Mn-oxide burial in

environments with $O_2 < 10\,\mu M$, likely under high $Mn^{2+}$ seawater conditions, should see a decrease in $\varepsilon^{205}Tl_{sw}$ with little impact on $\delta^{51}V_{sw}$, as seen immediately after the first S-MIF disappearance in the lower section. Expansion of environments with dissolved $O_2 > 10\,\mu M$, unaccompanied by an increase in Mn-oxide burial due to a limited seawater $Mn^{2+}$-reservoir, should see an increase in $\delta^{51}V_{sw}$ with little impact on $\varepsilon^{205}Tl_{sw}$, as seen in the shift from the lower to upper section. Error bars for $\delta^{51}V_{auth}$ are 2 SD of reproducibility on either the individual sample or the BDH chemicals V solution standard, whichever is larger. Error bars on previously published $\varepsilon^{205}Tl$ data[16] are 2 SD of reproducibility on either the individual sample or the SCo-1 geostandard run alongside samples, whichever is larger.

the isotopic difference between sediments and seawater would decrease[17]. By mass balance, as global marine V sinks become more oxidizing and thus isotopically lighter, $\delta^{51}V_{sw}$ should shift to more positive values. There are two end-member scenarios for how ocean oxidation could have driven a positive shift in $\delta^{51}V_{sw}$: one where there was no oxic V sink, and the positive shift resulted from more fractionated, non-quantitative vanadyl drawdown in reducing environments that became slightly more oxidized after ca. 2.32 Ga; and one where true oxic ($O_2 > 10\,\mu M$) V sinks appeared or expanded in the oceans after ca. 2.32 Ga.

To explain the full magnitude of the $\delta^{51}V_{sw}$ shift solely with changes to the reducing sink would require a $\geq 19\%$ decrease in the fraction of vanadyl drawdown with sinking organic matter, according to a Rayleigh distillation model (see "Methods", Fig. S2). With no other changes to the V cycle, this drawdown might be expected to scale with organic particle burial, requiring a concurrent $\geq 19\%$ drop in the fraction of organic carbon burial in the global oceans. Using a simple, two-component carbon isotope mass balance, such a change in the $C_{org}$ burial fraction could have induced a $> 4\%$ drop in seawater $\delta^{13}C$ (Fig. S2) that is not present in syn-GOE carbonate records. Even allowing for some degree of nonlinearity between carbon burial and V drawdown, modifications to the reducing V sink alone appear incapable of explaining the shift in $\delta^{51}V_{sw}$ during the GOE, without creating greater problems for the interpretation of traditional geochemical proxies.

Alternatively, an expansion of oxygenated ($> 10\,\mu M$ $O_2$) environments, driving greater stabilization of vanadate and its removal alongside various Fe-Mn oxyhydroxide-bearing sediments, would have established a highly fractionating sink with $\Delta^{51}V_{O2 > 10\,\mu M}$ around $-1.1\%$. This could have had a large impact on marine V isotope mass balance without requiring an overhaul of the global ocean carbon cycle, with a 0.27‰ positive shift in $\delta^{51}V_{sw}$ being accommodated with burial of a modest 25% of marine V in oxidized sediments (see Methods). For comparison, modern oxidized V burial on continental shelves and abyssal plains accounts for ~72% of the global sedimentary V sink[20]. A combination of the two end-member processes described above could provide the most parsimonious explanation for post-ca. 2.32 Ga

increase in $\delta^{51}V_{sw}$, because oxidation of the ocean interior to drive a decrease in organic carbon burial would likely require a top-down influence from shallow waters in communication with the newly oxygenated atmosphere. Critically, anything except the no-oxic-sink end-member model would require that true oxygenated ocean environments were globally established after ca. 2.32 Ga.

The development of any globally detectable oxidized V sink requires widespread marine environments with dissolved $O_2$ of $>10\,\mu M$ in bottom waters[17]. These oxygenated environments would most likely have been located at shallow water depths, assuming equilibration with rising atmospheric $pO_2$ alongside independent evidence for pervasive deep-ocean anoxia[33]. If the atmosphere provided this $O_2$ source in the aftermath of S-MIF disappearance, a Henry's Law calculation for $O_2$ solubility at 25 °C in seawater points to $pO_2 > 8 \times 10^{-3}$ atm or $4 \times 10^{-2}$ PAL, although intrinsic generation of some $O_2$ by cyanobacteria may also have contributed to this overall concentration. Box modeling studies indicate that a large volume of the ocean interior could have remained functionally anoxic ($O_2 < 1\,nM$) beneath the atmospheric $pO_2$ calculated above, provided that the biological pump operated at $\geq 20\%$ of its modern capacity[5]. A configuration of substantial top-down oxygenation, mostly restricted to the surface ocean, with oxidized V burial mostly developed on continental shelves bathed in these shallow waters, would also explain a smaller fractional oxidized V sink compared to the modern oceans.

A complementary constraint on the marine response to the GOE is provided by the appearance of light authigenic thallium (Tl) isotopic compositions ($\varepsilon^{205}Tl_{auth}$) in the oldest samples lacking S-MIF ($\Delta^{33}S = 0.0 \pm 0.3\%$) at the Rooihoogte-Timeball Hill formation boundary (Figs. 1F, 4A)[16]. These data provide evidence for geologically rapid shallow-ocean equilibration with an oxygenated atmosphere and burial of high-$\varepsilon^{205}Tl$ Mn oxides, producing a complementary low seawater $\varepsilon^{205}Tl$ ($\varepsilon^{205}Tl_{sw}$) value[16]. Younger Timeball Hill Formation strata lack such low $\varepsilon^{205}Tl_{auth}$, despite evidence for an at least intermittently oxygenated atmosphere on the basis of the S-MIF record (Fig. 1)[16]. Because Tl isotope fractionation is driven specifically by Mn oxide burial[18], rather than a particular abundance of dissolved $O_2$, it is unclear from Tl isotopic data alone whether post-ca. 2.32 Ga, near-crustal

$\varepsilon^{205}Tl_{auth}$ values necessitate a period of relative ocean deoxygenation. This is because globally significant Mn oxide burial requires a sufficiently high combined product of dissolved $Mn^{2+}$ and $O_2$, such that an attenuation of post-ca. 2.32 Ga Mn oxide burial could be explained by either a decline in the availability of $O_2$ or the significant drawdown of an originally large pre-GOE dissolved marine $Mn^{2+}$ reservoir[16].

The new V isotopic data presented here, combined with existing Tl isotopic data, provide more clarity on the post-ca. 2.32 Ga oxygenation of the oceans, due to nuances in how the oxic sinks for each of these elements operate. Unlike the oxic Tl sink that is Mn oxide specific, the operation of a highly fractionated oxic V sink is dependent on the stabilization of vanadate at dissolved $O_2$ levels >10 μM, which is then adsorbed onto a diffuse global flux of various Mn and (dominantly) Fe oxyhydroxides[17,19]. This makes $\delta^{51}V_{sw}$ less sensitive to lower dissolved $O_2$ levels that may still have promoted major Mn oxide burial, and $\varepsilon^{205}Tl_{sw}$ perturbations, in the $Mn^{2+}$-rich pre/syn GOE oceans[50]. However, $\delta^{51}V_{sw}$ responses to ocean oxygenation above 10 μM $O_2$ should have been less impacted by the specifics of the marine Mn cycle in the aftermath of atmospheric oxygenation[16,51]. Therefore, the combination of $\varepsilon^{205}Tl$ records, which have more expansive seawater archives, but more specific redox drivers, and $\delta^{51}V$ records, which may respond directly to marine $O_2$, but have more complex, only qualitative seawater archives and a higher oxygenation threshold, provides more texture to the history of early ocean oxygenation than either proxy can alone.

Vanadium isotope data suggest that post-ca. 2.32 Ga shallow marine bottom water $O_2$ levels may have exceeded 10 μM by the time of deposition of the upper section, so it is unlikely that Mn oxide burial would have been limited by $O_2$ availability. Therefore, the more likely interpretation of the Tl isotope record at this time is that extensive earlier Mn oxide burial had drawn down a large pre-GOE seawater $Mn^{2+}$ reservoir to such a degree that Mn oxide burial became limited by $Mn^{2+}$ availability[16]. Indeed, a possible negative correlation developed by small $\delta^{51}V_{auth}$ and $\varepsilon^{205}Tl_{auth}$ variations within the upper section may reflect the diminished 'response signal' of $\varepsilon^{205}Tl_{sw}$ to small fluctuations in marine $O_2$ in the face of a diminished $Mn^{2+}$ pool from which to form Mn oxides (Fig. 4). An alternative scenario of rapid, post-ca. 2.32 Ga deoxygenation, driven by oxidative weathering and eutrophication, has been suggested based on a multiparameter local redox dataset covering the upper Rooihoogte and lower Timeball Hill formations[52]. The lower Timeball Hill Formation documents oxic local redox conditions in the EBA-2 drill core that were unsuitable for $^{51}V_{auth}$ measurements, leaving the door open for that alternative scenario to have operated on a short geological timescale. Regardless, within the upper Timeball Hill Formation, deoxygenation does not seem to remain a viable explanation for near-crustal $\varepsilon^{205}Tl_{auth}$ values, because this is the interval where we infer the positive shift in $\delta^{51}V_{sw}$.

Multiproxy evidence now allows unprecedented reconstruction of the marine response to atmospheric oxygenation after ca. 2.32 Ga (Fig. 5). Coincident with S-MIF disappearance at ca. 2.32 Ga across the Rooihoogte-Timeball Hill formation boundary, atmospheric $O_2$ established shallow-marine $O_2$ concentrations sufficient to drive Mn oxide burial in $Mn^{2+}$-rich oceans, producing fractionated $\varepsilon^{205}Tl_{sw}$ values alongside S-MIF disappearance[16]. Global $\delta^{51}V_{sw}$ remained initially unaffected throughout this interval, suggesting dissolved bottom water $O_2$ remained <10 μM in most of the shallow ocean (Fig. 5). Loss of fractionated $\varepsilon^{205}Tl_{sw}$ values after ca. 2.32 Ga suggests some attenuation of a large marine Mn reservoir (Figs. 4, 5). Subsequently, the surface ocean equilibrated with atmospheric $O_2$, maintaining the shallow water column at levels >10 μM and establishing a globally impactful oxidized V sink for the first time. While numerous geochemical proxy records agree that extensive deep-ocean anoxia existed at this time, and persisted for at least another 1.5 Gyr[4,53], empirical evidence now suggests that Earth's first rise of atmospheric oxygen was globally propagated into the underlying shallow oceans on a timescale that was short compared to its planetary lifetime.

## Methods

### Geological setting of samples

Samples were obtained from the EBA-2 diamond drill core (26.4700° S, 27.5883° E) drilled near Carltonville, South Africa (Kloof Goldfields Property, Eastern Boundary Area). This drill core intersected a well-preserved interval of the Paleoproterozoic Transvaal Supergroup, and the specific depth intervals analyzed in this study have been sampled and analyzed in several previous studies[2,3,12,16]. The interval of the EBA-2 drill core we studied comprises the Rooihoogte and Timeball Hill formations. The succession has undergone regional metamorphism to only lower greenschist facies grade[54].

The Rooihoogte Formation has upper and lower members in the drill core EBA-2, and only the upper Rooihoogte Formation was analyzed and is thus described in this study. It dominantly consists of mudstones and black shales that coarsen upwards into siltstone and are capped by a thin chert breccia[42]. The Timeball Hill Formation contains two upward-coarsening sequences. In each sequence, the lower part is highly carbonaceous black mudstone, and these mudstones become less carbonaceous up-section, where they are interbedded with dark-gray to gray siltstones[42]. The mudstones and black shales of the upper Rooihoogte and Timeball Hill formations are interpreted to have been deposited in a pro-delta setting within the basin, with a connection to the open ocean towards the southwest[55,56]. Diamictite and conglomerate of the Rietfontein Member form a cap for the upper Timeball Hill Formation[42]. The Rietfontein diamictite has been interpreted to be a glacial deposit based on the presence of faceted and striated pebbles[57], and was not sampled for authigenic V isotopic analysis in our study.

### Vanadium isotope analysis

Samples were prepared for V isotope analysis in a Class 100 clean laboratory at Woods Hole Oceanographic Institution (WHOI). All reagents used were double-distilled in the laboratory or purchased at trace-metal grade or higher. Approximately 200 mg of powdered sample material was weighed into acid-cleaned Savillex Teflon vials and leached overnight in 2 M $HNO_3$ at 130 °C, following a protocol that is shown in numerous previous studies to isolate authigenic V[17,40]. The leached fraction was then separated by centrifugation and pipetting of the supernatant into cleaned Teflon vials for multiple further acid digestion steps. Acid digestion steps included the use of concentrated $HCl + HNO_3$ (aqua regia) and concentrated $HNO_3 + H_2O_2$, which destroy organics, but avoid digesting any detrital silicate material. Aliquots of these exact same leach solutions were utilized for previous Tl isotope analyses[16]. Aliquots of the samples were then brought into solution in 2 mL of 0.8 M $HNO_3$ and processed through a four-step ion-exchange column chromatography procedure to purify V from matrix elements, particularly Ti and Cr, which have isobaric interferences with V[58,59]. Load and matrix fractions collected for each column were analyzed using a Thermo Fisher iCAP quadrupole inductively coupled plasma mass-spectrometer (Q-ICPMS) at WHOI to determine that no V was lost during column purification.

Vanadium isotope ratios were measured on a Thermo Neptune multi-collector ICP-MS (MC-ICP-MS) at WHOI following established methods[60]. Both samples and standards were prepared to matched concentrations of 800 ng/ml in 0.1 M $HNO_3$. Sample introduction was conducted using a Cetac Aridus II desolvating nebulizer, and in the Neptune, nickel 'Jet' type sampler and 'X' type skimmer cones. Analyses were conducted in medium resolution mode to resolve flat-topped V (and Cr, Ti) peak shoulders away from various $ArC^+$ and $ArN^+$ (plus hydride) interferences. In this configuration, we typically achieved 150–250 V/800 ppb V sensitivities on $^{51}V$. To correct for isobaric

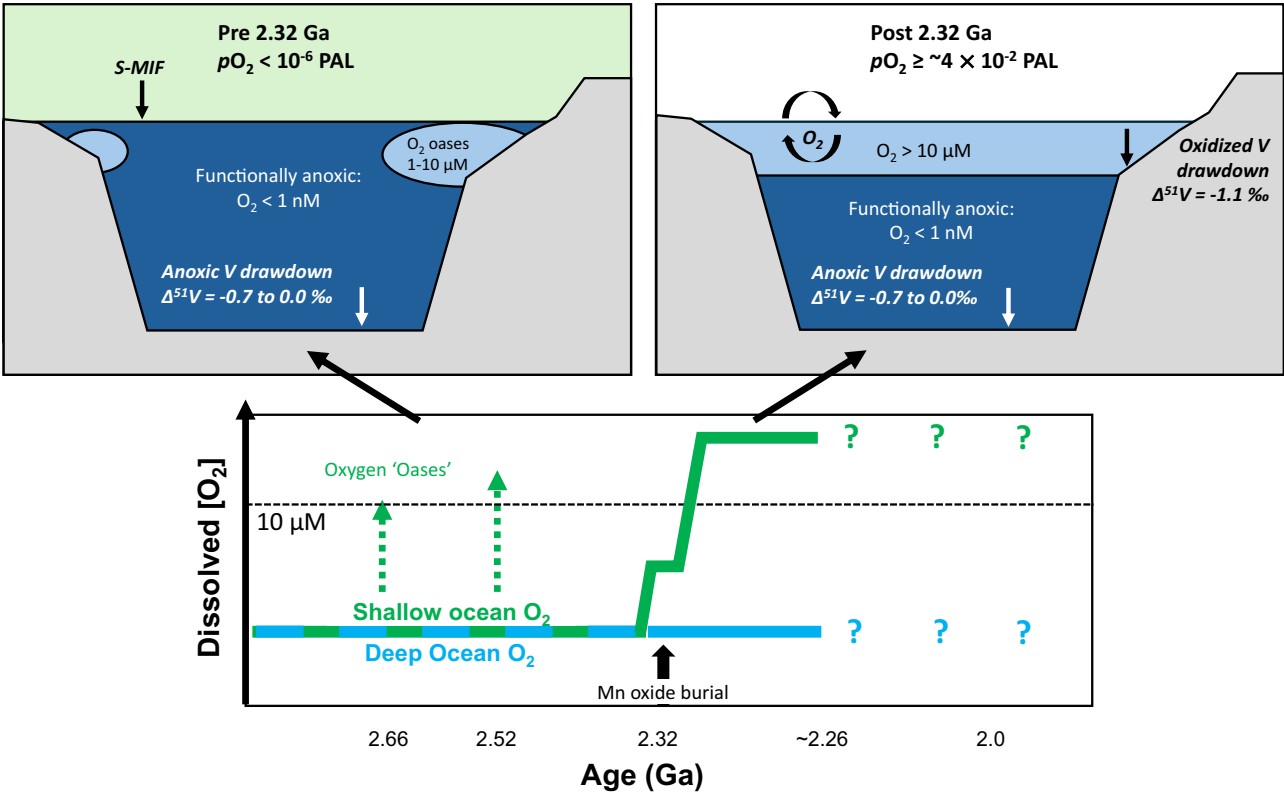

**Fig. 5 | Schematic illustration of inferred atmosphere-ocean oxygenation state before and after ca. 2.32 Ga informed by V isotope composition.** Prior to ca. 2.32 Ga, during deposition of the upper Rooihoogte and lower Timeball Hill formations, atmospheric $pO_2 < 10^{-6}$ PAL facilitated sulfur isotope mass-independent fractionation (S-MIF) signatures and a global ocean dominated by anoxic conditions and strong anoxic V drawdown. Any $O_2$ oases that were present did not strongly impact V isotope mass balance. At ca. 2.32 Ga, atmospheric $pO_2$ rose above $10^{-6}$ PAL, driving a disappearance of S-MIF and onset of extensive Mn-oxide burial, while bottom-water dissolved $O_2$ initially remained below 10 µM in shallow marine environments. Subsequently, by the time of deposition of the upper Timeball Hill Formation, surface ocean redox conditions reached $O_2 > 10$ µM, reflecting equilibration with $pO_2 \geq$ ca. $4 \times 10^{-2}$ PAL, establishing a persistent, oxidized V sink on continental shelves. While most of the seafloor remained functionally anoxic with strong V drawdown, the new oxic sink increased the seawater V isotope composition by $\geq 0.27$‰. In the bottom panel, the blue and green lines indicate marine dissolved $O_2$ in the deep and shallow oceans, respectively, while the black arrow marks the onset of major Mn oxide burial in response to rising $O_2$ recorded by Tl isotopes[16].

interference from minor contaminant $^{50}$Cr and $^{51}$Ti on the V isotope signals, Cr and Ti isotopes were analyzed simultaneously with $^{50}$V and $^{51}$V and isobar contributions were corrected for using a mass bias coefficient that was determined at the end of the analytical session by analysis of a pure Cr plasma tuning solution. All isotopes of interest were measured on Faraday cups fitted with $10^{11}$ Ω resistors, except $^{51}$V, which was measured using a $10^{10}$ Ω resistor. Analysis used standard-sample bracketing with the Alfa Aesar (AA) V specpure standard solution to correct for instrumental mass fractionation, with V isotopic ratios reported in delta notation: $(\delta^{51}V$ (‰) $= 1000 \times \{[(^{51}V/^{50}V)_{sample} - (^{51}V/^{50}V)_{AA}]/(^{51}V/^{50}V)_{AA}\})$. Each sample was bracketed with analysis of the BDH Chemicals V solution internal standard with known $\delta^{51}V^{17,35,58}$ that was analyzed identically to samples to monitor instrument performance, giving values in agreement with the published $\delta^{51}V$ value of −1.19‰ over each full analytical session $(\delta^{51}V = -1.18 \pm 0.08$‰ (2 SD) in session 1; $\delta^{51}V = -1.21 \pm 0.15$‰ (2 SD) in session 2). Samples were analyzed in duplicate or triplicate, according to the amount of available material, and uncertainties are presented as the larger of either the 2 SD uncertainty of BDH analyses run in the same sequences or the 2 SD of replicate analyses of the samples. Bulk digests of the BHVO-2 and AGV-2 USGS igneous rock geostandards, and an authigenic leach of the SCo-1 USGS shale geostandard, were processed through the column chromatography and mass spectrometry protocols alongside drill-core EBA-2 sedimentary rock samples and yielded values in agreement with those previously published[17,19,59,61,62].

Our V isotope dataset was generated by analysis of two separate batches of samples over separate analytical sessions (Supplementary Data File). We saw no evidence for systematic differences in $\delta^{51}V$ values measured in each session. Of the geostandards measured in both sessions (SCo-1 leach, BHVO-2), one was lighter (but within error) and another was equal to or heavier (but within error) than the recommended value for each session, and the BDH values were almost identical (second session slightly lighter), and always within error of the recommended value. For the EBA-2 drill-core shale samples, no upper section samples (<1300 m depth) were analyzed in the first session. However, the average value of the lower section (>1300 m) samples was identical in each session (first session: $\delta^{51}V_{auth} = -1.07 \pm 0.08$ 2SE, second session: $\delta^{51}V_{auth} = -1.07 \pm 0.11$ 2SE). Therefore, the positive $\delta^{51}V$ shift (averaging 0.27‰) observed from the lower to upper section appears not to be an artifact of different analytical sessions.

**Trace element concentrations**

Bulk and authigenic V and a suite of other trace-element concentrations were determined from a separate bulk rock digest, and an aliquot of the authigenic leachate produced for V isotope analysis, respectively. These concentration measurements were made during the previous Tl isotopic study, and the methods used for these analyses are briefly outlined below for ease of reference. Bulk digestion of ~20 mg of rock powder was performed using concentrated acid steps,

including HF + $HNO_3$, HCL + $HNO_3$, and $HNO_3$ + $H_2O_2$. For both bulk and leachate material, elemental concentrations were generated using a Q-ICP-MS analyses at WHOI with reference to a five-point calibration curve based on dilutions of a gravimetrically prepared, multi-element standard. Prior to analysis, samples were diluted in 2% $HNO_3$ and doped with indium (In) to act as an internal standard for monitoring matrix effects and instrumental drift. Trace-element concentration measurements performed this way at WHOI have been shown to be accurate and have a precision of 5–10% depending on individual elements, based on comparison to replicate analyses of USGS reference materials AGV-1, AGV-2, BHVO-1, BHVO-2, BIR-1, and BCR-2[63,64].

**Vanadium isotope fractionation calculations**

We do not attempt to pinpoint the exact values of $\delta^{51}V_{sw}$, or the globally averaged effective $\Delta^{51}V$ that prevailed during deposition of the Rooihoogte and Timeball Hill formations, due to limited constraints in the local fractionation of V into sediments. Two end-member scenarios are suggested to explain the relative change in $\delta^{51}V_{sw}$ of a + 0.27‰ between the average values inferred for the lower and upper sections. We considered two scenarios: one where there was no oxic V sink, and a positive shift in $\delta^{51}V_{sw}$ resulted from a shift to non-quantitative, fractionated vanadyl drawdown in reducing environments that became slightly more oxidized after 2.32 Ga and one where an oxic ($O_2 > 10 \mu M$) V sink appeared or expanded.

In the scenario that the increase in $\delta^{51}V_{sw}$ solely reflects a decrease in the extent of vanadyl drawdown within a single global reducing sink:

$$\delta^{51}V_{rivers, in} = \delta^{51}V_{sediments, out} \quad (1)$$

and

$$\delta^{51}V_{sediments, out} = \delta^{51}V_{SW} + \Delta^{51}V_{eff} \quad (2)$$

where $\Delta^{51}V_{eff}$ is the effective isotopic difference between seawater and sediment. In the extreme case that initially in the lower section, $\Delta^{51}V_{eff} = 0‰$ (full drawdown of vanadyl)[20], an increase of $\delta^{51}V_{sw}$ by +0.27‰ would need to be offset by a decrease of $\Delta^{51}V_{eff}$ to −0.27‰ during deposition of the upper section. If we treat this sink as the cumulative product of Rayleigh distillation with an instantaneous isotopic difference $\Delta^{51}V_{inst} = −0.7‰$ (as seen during V drawdown to the Cariaco Trench sediments)[17], then:

$$\Delta^{51}V_{eff} = \delta^{51}V_{cumulative} - \delta^{51}V_{initial, dissolved} = (f - 1) \times \Delta^{51}V_{inst} \times \ln[(f - 1)/f] \quad (3)$$

The smallest change in $f$ associated with a 0.27‰ decrease in $\Delta^{51}V_{eff}$ is found when $f$ goes from 1 to 0.81, giving a minimum 19% decrease in organic particle-associated V burial (Fig. S2). We note that this endmember scenario poses an unrealistic initial (lower section) condition of total, quantitative drawdown of vanadyl from seawater, but we use this endmember as it sets the minimum relative change in V drawdown burial that could have occurred going from the lower to upper section.

For the oxic sink scenario, we can again determine the minimum relative change in the size of the oxic sink using the endmember scenario that the whole shift in $\delta^{51}V_{sw}$ reflects a global change in $\Delta^{51}V_{eff}$ from 0‰ in the lower section, and −0.27‰ in the upper section. If we assume, in this case, that the entire 0.27‰ increase in $\delta^{51}V_{sw}$ solely reflects new V removal to an oxidized vanadate sink with $\Delta^{51}V_{O2 > 10 \mu M} = −1.1‰$[17], while the remainder of V is still removed to highly reducing sink with $\Delta^{51}V_{red} = 0‰$, then the increase in fraction of oxidized V burial can be found by solving:

$$\Delta^{51}V_{eff} = f_{ox} \times \Delta^{51}V_{O2 > 10 \mu M} + (1 - f_{ox}) \times \Delta^{51}V_{red} = -0.27‰ \quad (4)$$

setting $\Delta^{51}V_{red} = 0$, and rearranging to find that $f_{ox} = 0.27/1.1 = 25\%$, which is approximately 1/3 of the modern ocean oxidized V sink fraction[20].

## Data availability

All data reported in the present study are included as Supplementary Information in Supplementary Data File S1.

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

## Acknowledgments

This work was funded by NASA Exobiology grant 80NSSC22K1628 (C.M.O., A.W.H., S.G.N.), the WHOI postdoctoral scholar program (A.W.H., C.M.O.), the Agouron Institute Fellowship in Geobiology (A.W.H.), Discovery and Accelerator Grants from the Natural Sciences and Engineering Research Council of Canada (A.B.), ACS Petroleum Fund grant 624840ND2 (A.B.), and Natural Environmental Research Council grant NE/Z000122/1 (S.W.P.). Maureen Auro and Jerzy Blusztajn are thanked for laboratory and instrument support at WHOI.

## Author contributions

A.W.H. and S.G.N. conceptualized the study in collaboration with C.M.O. A.B., and S.W.P. collected and provided samples. A.W.H., C.M.O., and Y.S. prepared the samples for geochemical analysis. A.W.H. performed isotopic analysis. A.W.H. drafted the initial manuscript. C.M.O., A.B., S.W.P., and S.G.N. helped A.W.H. to revise the manuscript before submission.

## Competing interests

The authors declare no competing interests.
