## [Transparent Peer Review file · Nature Communications]

Onset of persistent surface ocean oxygenation during the Great Oxidation Event

Corresponding Author: Dr Andy Heard

Version 0:

Reviewer comments:

Reviewer #1

(Remarks to the Author)

This is my second time to review this paper. Considering this is a new submission to Nature Communications, I go through the whole MS as a new one. Of course, this is still a very important study, and the new V isotope dataset is with high quality and scientific significance. In addition, authors have addressed most of the points raised by the previous reviewers, and the MS has been certainly improved. I appreciate their effort to take the reviewers' comments into account, and suggest publication after considering my new comments listed below.

Line 37. Could you please introduce more details on the trajectory of biological innovation during the GOE?

Line 60. I encourage authors to consider a more important scientific question, rather than complementing the TI isotope perspective on ocean oxygenation across the GOE using V isotopes.

Line 74–77. But the residence time of oceanic V may be shorter than 90 kyr in ancient oceans.

Line 77–78. Deposition of dissolved V into sediments with different isotope fractionation should be involved in the reasons why V isotope geochemistry provides information on the global ocean redox state, which I think is the most important.

Line 86. I don't think this isotope fractionation is only controlled by Fe-oxyhydroxide.

Line 97–99. This should be indicated in Fig. 2.

Line 118. You should at least provide the location of this drill core.

Line 156. Referred to as the "upper section"?

Line 161. Again, you should make sure if the paleoseawater $\delta^{51}\text{V}$ value was globally homogeneous!

Line 165–166. I can see a robust positive correlation between the TOC and V EF, which could support the removal of oceanic V by organic matter.

Line 201–207. Maybe, some paleo-salinity proxies can help to eliminate this effect.

Line 221–226. Figures are necessary to visually illustrate the Rayleigh distillation model (Line 223) and two-component carbon isotope mass balance model, at least in Supplementary materials (Line 225–226).

Line 233–236. See the comment above.

Line 260. "These data provide".

Line 472–488. Figs. 1H-N are not described.

Figure 1. The relationship of (A-G) with (H-N) is not shown clearly. Authors can revise as Ostrander et al. (2024).

Figure 2. The V isotope fractionation between euxinic sediments and open-ocean seawater is from -0.7‰ to 0.0‰ , controlled by local V drawdown efficiency. Please revise it.

Wei Wei

University of Science and Technology of China

Reviewer #2

(Remarks to the Author)

I was reviewer #3 in the previous round of submission of this manuscript. In re-reviewing the manuscript here, I found that the responses to my queries were answered satisfactorily and that the overall clarity of the manuscript was substantially improved by the revisions suggested by all reviewers.

Reviewers Comments

Author responses

"Text from revised MS"

Reviewer #1 (Remarks to the Author):

This study builds on work done on the EBA-2 drill core from South Africa that goes from the Rooihoogte to Rietfontein formations, spanning age constraints of 2.32 to 2.26 Ga, with samples previously measured for S-MIF (Poulton et al., 2021) and TI isotopes (Ostrander et al., 2024). In this new study, V isotopes are measured that could offer additional insight on ocean anoxia and oxygenation during the time of the Great Oxygenation Event (GOE) and its aftermath. Although the use of $\delta^{51}\text{V}$ is interesting and novel, there are important limitations in this study.

We appreciate the reviewer's comment that the study is interesting and novel, and hope we can demonstrate below that we have both addressed the limitations noted, and explained how some of the perceived limitations stem from a slight misinterpretation of earlier work we have published.

It is conspicuous that the TI isotopes (Ostrander et al., 2024) and FeHR/FeT, which are interpreted as reflecting global marine water column redox, shift towards anoxia during the S-MIF oscillations (Poulton et al., 2021), but this is not reflected in the response of $\delta^{51}\text{V}$, which should respond to local conditions:

We should first note that we did not dwell heavily on the S-MIF reappearances high in the stratigraphy because these isolated signals are currently the subject of active debate. We also refer the reviewer to our supplementary discussion, "Reconciling vanadium isotope constraints on marine oxygenation with possible reappearances of S-MIF", which we copy below in its entirety for ease of access.

"Sporadic, short returns of S-MIF in the upper Timeball Hill Formation, potentially suggesting short-lived, atmospheric deoxygenation events between ~ 2.32 and 2.22 Ga^1 , are mostly observed in samples deemed unsuitable for V isotopic analyses due to oxic local redox conditions (Fig. 1). The origin of these younger returns of non-zero S-MIF are still heavily debated¹⁻³, and due to the unresolved nature of this debate and its peripheral connection to the data presented herein, they are only briefly discussed in connection to V isotopic data below. The $\delta^{51}\text{V}_{\text{auth}}$ dataset does include two isolated samples with non-zero S-MIF, at 1091 and 943 m depth. In each case, $\delta^{51}\text{V}_{\text{auth}}$ in these samples are indistinguishable from the upper Timeball Hill Formation samples lacking S-MIF (Fig. 1F), suggesting that $p\text{O}_2$ perturbations at

this time were not transferred to the oceans in a manner that impacted the marine V cycle. Two phenomena that may each fully or partially explain these observations concern the duration of any possible short-lived S-MIF returns, and the partitioning of V between different, seafloor redox sinks. Statistical approaches were recently used to show that the development of short S-MIF reappearances in the Timeball Hill Formation sections^{1,2} requires that any ephemeral returns to an anoxic atmospheric redox state would have lasted for as little as tens of thousands of years³. These timescales overlap with the modern (91 kyr) V residence time in seawater. Even accounting for a shorter V residence time under more anoxic ocean conditions associated with a smaller seawater V reservoir, it is unclear that such a short-lived perturbation in surface marine O₂ forcing would have propagated its effect to the global $\delta^{51}V_{sw}$ value. This is because global ocean geochemical responses to forcings are expected to be best expressed when forcings are longer in duration than the residence time of the element of interest⁴."

With respect to TI isotopes, a "shift towards anoxia during the S-MIF oscillations" is not a completely correct representation of the results of Ostrander et al. (2024), and it appears the reviewer may have slightly misinterpreted a takeaway of the earlier published TI isotope study. Higher in the Timeball Hill Fm we saw no major $\epsilon^{205}Ti_A$ shifts, even across the short-term S-MIF returns/disappearances – despite the schematic cartoon at the end of that paper showing how such oscillations might look. We have added the TI isotope data to Fig. 1 where it is evident that TI isotopes do not evolve to substantially sub-crustal values or otherwise resolvably oscillate with S-MIF higher in the section. We argued in our 2024 paper that it is a diminished dissolved Mn²⁺ reservoir that is preventing large negative $\epsilon^{205}Ti_A$ shifts in these higher units. The dependence specifically on Mn oxides for TI but not V isotopes (the latter of which are mostly impacted globally by Fe oxyhydroxide burial) explains the decoupling of these proxies and underlines why using multiple proxies provides more detail on ocean redox, giving our study more value and meaning than if it was simply a rehashing of previous conclusions using a different proxy. We have added additional discussion to the revised main text outlining how different mineral carriers of TI and V isotopic signals in global oxic sinks can help explain the decoupling of their reconstructed seawater signatures. Please refer to our responses to the specific queries of Reviewer #2 that are pertinent to this question.

FeHR*/FeT are indicators of changing local, not global, redox, and the changes this proxy documents provided some of the motivation for our extensive discussion (including Fig. 3) of different possible local-redox-dependent $\Delta^{51}V$ values to use when reconstructing global seawater $\delta^{51}V$. Therefore, this comment is confusing to us, because we made a concerted

effort to try and account, within acknowledged uncertainties, for the difficulties in reconstructing global ocean $\delta^{51}\text{V}$ in the face of variable local redox. We should also emphasize that because we tried to target anoxic samples wherever possible, the full range of local FeHR/FeT variation seen in the study of Poulton et al. is not represented in and does not impact the samples we analyzed.

Line 68-69: "Dissolved V is deposited in sediments with isotopic differences relative to seawater ($\Delta^{51}\text{V} = \delta^{51}\text{V}_{\text{sediment}} - \delta^{51}\text{V}_{\text{sw}}$) that are controlled by local redox conditions^{16,17}."

The central claim that V isotopes show "unidirectional transition in global ocean redox conditions" from 2.32 to 2.26 Ga is not fully convincing. It appears there is some nuance here that is not addressed or developed. Why do S-MIF and TI isotopes show "oscillations" interpreted for anoxic atmosphere but V isotopes that should respond more locally do not? Please refer to our response above for more detail with regards to S-MIF reappearances and TI isotopes. Again, TI isotopes only strongly covary with S-MIF at the canonical S-MIF disappearance interval, while higher in the Timeball Hill Fm we see little variability in $\epsilon^{205}\text{Ti}_A$, even during the short-term S-MIF returns, likely due to limitations in shallow water Mn supply, rather than O_2 . We have also added extensive discussion near the end of the main text (and Fig. 4), to explain why and how TI and V isotope records may or may not be coupled to one another, due to the different nature of their oxic sinks. The benefit of using multiple paleoredox proxies on the same section is that we can use their different nuances to glean more information about Earth history than any one proxy alone. The goal was never to simply replicate the TI isotope conclusions with a new proxy approach.

The sparse S-MIF reappearances higher in the stratigraphy are themselves under active debate right now, which is why we did not anchor a large part of our discussion to a few contentious datapoints, and discussed those S-MIF reappearances in the Supplementary Information where it would not break the flow of the main text.

As atmospheric oxygenation is a global phenomenon, the intent of reviewer's comment about local responses is unclear. Regardless, our reconstructed seawater V isotopic interpretations reflect our best estimate after correcting for local influences during depositions to recover the *global* (homogeneous) seawater value and thus discuss the global ocean redox state. We have added additional text to the V isotope introduction to make the point clear that V isotopes can be utilized as a global, not a local, proxy.

"Vanadium isotope geochemistry provides information on the global ocean redox state because 1) V is redox sensitive, behaving differently in different redox environments (Fig. 2); and 2) it has a long (ca. 90 kyr) residence time relative to modern and ancient ocean mixing

timescales on the order of 1 kyr²⁰, such that the dissolved V reservoir and its isotopic signature should be globally well-mixed in the open ocean and unrestricted basins¹⁹. Because of this, reconstruction of marine V isotope mass balance from sedimentary archives can shed light on the ancient ocean redox state."

Line 49-51: "The significance of younger S-MIF is still debated (Supplementary Information), but multiple possible fluctuations across the GOE interval suggest that atmospheric pO₂ may have oscillated above and below 10⁻⁶ PAL during a transition lasting from ~2.43 to 2.22 Ga^{1,10-12}."

Then, there is some limited discussion relegated to the supplemental:

"The $\delta^{51}\text{V}_{\text{auth}}$ dataset does include two isolated samples with non-zero S MIF, at 1091 and 943 m depth. In each case, $\delta^{51}\text{V}_{\text{auth}}$ in these samples are indistinguishable from the upper Timeball Hill Formation samples lacking S-MIF (Fig. 1F), suggesting that pO₂ perturbations at this time were not transferred to the oceans in a manner that impacted the marine V cycle."

Why should a ostensibly global marine redox proxy like Tl isotopes respond to the apparent pO₂ changes (S-MIF oscillations younger than 2.32 Ga) but an apparently more local ocean redox proxy, $\delta^{51}\text{V}$, not respond?

Again, Tl isotopes uniformly return to crustal values after the canonical S-MIF disappearance around 2.32 Ga, they do not show an oscillation in the higher stratigraphy in the manner that the reviewer claims based on their misinterpretation of the Ostrander 2024 study. We reiterate that original interpretation of post-2.32 Ga Tl isotope systematics also invoked a role for the Mn cycle, to which V isotopes should not necessarily have the same sensitivity. Also, to repeat our point above, after the corrections for local fractionation effects that we discuss in detail in our manuscript, the seawater V isotopic compositions we discuss should also be *global* signal. An isotopic proxy is not inherently a local redox tracer just because local effects require correcting for. This should not be an unfamiliar concept given that similar considerations exist for the more established Mo isotopic system, and the unfractionated archive offered by Tl isotopes in appropriate shale samples is a fortuitous exception, rather than the norm.

All within the same core, EBA-2. This is very puzzling, and further that there is not more discussion about this.

We do not understand the reviewer's claim that 'there is not more discussion'. There is more discussion about this, in the following text of the same paragraph from the supplement that the reviewer just quoted above. We made a concerted effort to provide an explanation for these two datapoints, even though, as we emphasize again, the S-MIF in those samples is currently heavily debated by other workers in the S-MIF community.

This could be a very important result but here it appears as an important shortcoming of this study, instead, because a more in depth interpretation of this is avoided.

It was not avoided. Here is our 'in depth interpretation', copied from the very same supplemental text.

"Two phenomena that may each fully or partially explain these observations concern the duration of any possible short-lived S-MIF returns, and the partitioning of V between different, seafloor redox sinks. Statistical approaches were recently used to show that the development of short S-MIF reappearances in the Timeball Hill Formation sections^{1,2} requires that any ephemeral returns to an anoxic atmospheric redox state would have lasted for as little as tens of thousands of years³. These timescales overlap with the modern (91 kyr) V residence time in seawater. Even accounting for a shorter V residence time under more anoxic ocean conditions associated with a smaller seawater V reservoir, it is unclear that such a short-lived perturbation in surface marine O₂ forcing would have propagated its effect to the global $\delta^{51}V_{sw}$ value. This is because global ocean geochemical responses to forcings are expected to be best expressed when forcings are longer in duration than the residence time of the element of interest⁴."

Critically, it is problematic that V isotopes were not measured for the interval just above the 2.32 Ga horizon, depth 1300 to 1100 meters in the core (EBA-2), with scant explanation of why the "definitively oxic" sediments were not measured for $\delta^{51}V$, and this deserves elaboration.

Line 110-112: "We avoided intervals of between ~1100 and 1300 m depth that were deposited under definitively oxic conditions, due to a lack of precedent for reconstructing $\delta^{51}V_{sw}$ from such sediments."

Line 116: "The targeted samples were most likely deposited under reducing conditions."

Line 480-482: "We do not attempt to pinpoint the exact values of $\delta^{51}V_{sw}$, or the globally averaged effective $\Delta^{51}V$ to prevailed during deposition of the Rooihogte and Timeball Hill

formations due limited constraints in the local fractionation of V into sediments."

There is an unsatisfying circularity to not measuring the definitively oxic sediments because of not being able to easily reconstruct seawater $\delta^{51}\text{V}_{\text{sw}}$, then stating the targeted samples were deposited under reducing conditions (reduced sediments were more or less targeted), then that reconstruction of $\delta^{51}\text{V}_{\text{sw}}$ is anyhow not exact. The transition from more anoxic to oxic should occur within the unmeasured interval. Even if the reconstruction of seawater $\delta^{51}\text{V}$ is difficult for the 1300-1100 m interval in the core, surely this is one of the most important intervals to measure. At the very very least, the reader deserves further elaboration on why this interval was not measured.

We would, of course, have preferred to work with a succession that provided us an uninterrupted archive of seawater V isotope systematics across this interval that we agree with the reviewer is important, and would allow us to more precisely constrain the timing of the seawater V isotopic shift. However, no such archive at this very specific stratigraphy that preserves the canonical disappearance of S-MIF was available to us.

Of the samples between 1300 and 1100 m depth that were available to us, every single one featured iron speciation values ($\text{Fe}_{\text{HR}}^*/\text{Fe}_{\text{T}} < 0.22$) that place them in the oxic environmental zone and featured V_{EF} values close to 1, indicating limited authigenic V enrichment as seen in modern oxic settings. In oxidized environments, V is deposited adsorbed to the surfaces of Fe and Mn (oxyhydr)oxides which should be attacked with different chemical leaches to those applied for reducing sediments where V(IV) is bound to organic matter. However, the authigenic V leaching procedure used in modern oxic sediment V isotopic analysis has not been validated in sedimentary rocks. This contrasts with leaches for reducing sediments which have been calibrated in shale standards and modern sediments and have precedent of being applied to other studies of ancient sediments.

There is thus no currently established precedent in previous V isotope paleoredox studies for reconstructing seawater values from explicitly oxic sediments, and it would be bad practice to break this specifically for the application to 2.3-billion-year-old rocks that occupy a remarkable place in Earth history. There are greater potential pitfalls in applying a proxy to samples where we would not trust our own data and methodology, than there are in having a less-than-ideal stratigraphic gap in our dataset that decreases the time resolution of our interpretation.

We have explained this to the reader in more detail in the revised text, as they might have the same questions about this important interval.

"Despite capturing an important interval of Earth history in the immediate aftermath of the first S-MIF disappearance, we did not analyze authigenic V isotopes in the stratigraphic

interval between ~1,100 and 1,300 m depth. All available samples from this interval were likely deposited under oxic conditions based on Fe_{HR^}/Fe_T ratios that are lower than 0.22 (Fig. 1B). This sedimentary redox condition requires different leaching procedures due to a different dominant host phase for authigenic V (vanadate adsorbed to Fe-oxyhydroxides), and this leaching procedure has yet to be applied to or calibrated for ancient sedimentary rocks, having only been tested in modern marine sediments¹⁶.”*

As for the samples we did target, we took extensive efforts to communicate the uncertainties inherent to reconstructing seawater $\delta^{51}V$ values even from reducing sediments where there is precedent for the application of leaches to ancient rocks. These inherent uncertainties are why we choose not to present single ' $\delta^{51}V_{sw}$ ' values for a given sample or section, and instead present the ranges shown in (now) Fig. 3. The logic we present in the main text, specifically, how we anticipate any $\Delta^{51}V$ correction to be of equal or larger magnitude in the upper section based on local redox, makes our qualitative conclusion of a positive shift in $\delta^{51}V_{sw}$ robust. We empathize with the reviewer's view that the limitations of the available samples and information they contain does not tie up all the questions one might have about this interval in Earth history. However, such frustrations are common when working with samples from this deep in Earth history and do not undermine what we are still able to glean from the data.

The last paragraph seems to want to tie everything up into a neat story, but given the above, there seems to be important nuance here that needs to be addressed to help explain why S-MIF and Tl isotopes are interpreted as having a more direct response to pO₂ changes (oscillations) but $\delta^{51}V$ does not.

We again reiterate – the Ostrander et al. (2024) paper did not report higher stratigraphic oscillations of Tl isotopes with returning S-MIF. The only place where Tl isotopes do notably covary with S-MIF while V isotopes do not, is at the canonical S-MIF disappearance interval, and our explanation for the lack of a V isotope response at this time is a key detail of the concluding paragraph. Thallium isotope variation is generally more muted in the upper section and where it does occur to a limited extent, there is actually evidence for a covariation with V isotopes (see new Fig. 4). We have added more detail to our discussion here to explain the nuances of Tl and V isotope responses to global oxygenation (see below in responses to Reviewer #2) and how they may reflect different dominant mineral drivers at different O₂ thresholds for their oxic sinks, and the new Fig. 4 should also be instructive in this regard.

Lastly, it could make the V isotope systematics easier to follow for the reader if they are introduced from source to sink, going from rivers-ocean-sediment, with perhaps a figure to illustrate how $\delta^{51}\text{V}$ is transferred and fractionated, and incorporated into different phases before preservation.

This is a great suggestion, and with the expanded space allowed by Nature Comms, we are able to provide just such a figure. It is now Fig. 2 in the revised text.

Reviewer #2 (Remarks to the Author):

Sorry for the delay to submit the review report. I have gone through the whole MS carefully, and I am happy to see that the new V isotope dataset has provided individual evidence for marine oxygenation during the GOE. The manuscript is well written and organized, and thus I suggest publication after minor to moderate revision.

We appreciate the reviewer's positive feedback and are pleased that they recommended publication after revision.

Line 61. What is the advantage of V isotopes as a paleo-redox proxy relative to Tl isotopes? Can V isotopes solve the problem on the ocean oxygenation during the GOE, which Tl isotopes cannot solve? I think authors should clarify it, rather than just providing an independent perspective.

This is a good point, we should have provided more up-front justification for the approach. We have provided additional text in the introduction, and in the concluding discussion, to further explain the different and specific strengths of each proxy, what phases or conditions their oxic sinks respond to, and the benefits of combining both proxies to build a nuanced and textured picture of ocean oxygenation. The new Fig. 4 should also help show the strength of combining these proxies, and different processes they track/information they convey.

From the revised Introduction:

"To complement the Tl isotopic perspective on ocean oxygenation across the GOE, we measured sedimentary V isotope ratios (reported as $\delta^{51}\text{V} = (\frac{^{51}\text{V}_{\text{sample}}}{^{51}\text{V}_{\text{AA Specpure}}} - 1) \times 1,000$) in the same Rooihogte and Timeball Hill shale samples previously analyzed for Tl isotope values¹⁵. Vanadium isotopes track the global marine redox state as Tl isotope values do, but the oxidized sink for V in the oceans records a threshold dissolved O_2 level¹⁶ ($> 10 \mu\text{M}$; outlined in detail below), rather than the specific burial of Mn oxides¹⁷. As such, combined V and Tl isotopic data can provide more nuance to reconstructions of global ocean oxygenation events and their impacts on multiple redox-sensitive element cycles¹⁸. In this study we targeted

organic-rich shales and analyzed $\delta^{51}\text{V}$ in the authigenic V fraction ($\delta^{51}\text{V}_{\text{auth}}$). This fraction represents the V scavenged from Paleoproterozoic seawater by sinking organic matter, the isotopic composition of which allows reconstruction of relative changes in the global ocean redox state¹⁹.”

From the end of the discussion, where we added extensive new text on the topic of Tl and V oxic sinks and their possible covariations, or lack of:

“A complementary constraint on the marine response to the GOE is provided by the appearance of light authigenic thallium (Tl) isotopic ratios ($\epsilon^{205}\text{Tl}_{\text{auth}}$) in the oldest samples lacking S-MIF ($\Delta^{33}\text{S} = 0.0 \pm 0.3\text{‰}$) at the Rooihogte-Timeball Hill formation boundary (Figs. 1F, 4A)¹⁵. This data provides evidence for geologically rapid shallow-ocean equilibration with an oxygenated atmosphere and burial of high- $\epsilon^{205}\text{Tl}$ Mn oxides, producing a complementary low seawater $\epsilon^{205}\text{Tl}$ ($\epsilon^{205}\text{Tl}_{\text{sw}}$) value¹⁵. Younger Timeball Hill Formation strata lack such low $\epsilon^{205}\text{Tl}_{\text{auth}}$, despite evidence for an at least intermittently oxygenated atmosphere on the basis of the S-MIF record (Fig. 1)¹⁵. Because Tl isotope fractionation is driven specifically by Mn oxide burial¹⁷, rather than a particular abundance of dissolved O_2 , it is unclear from Tl isotopic data alone whether post-ca. 2.32 Ga, near-crustal $\epsilon^{205}\text{Tl}_{\text{auth}}$ values necessitate a period of relative ocean deoxygenation. This is because globally significant Mn oxide burial requires a sufficiently high combined product of dissolved Mn^{2+} and O_2 , such that an attenuation of post-ca. 2.32 Ga Mn oxide burial could be explained by either a decline in the availability of O_2 or the significant drawdown of an originally large pre-GOE dissolved marine Mn^{2+} reservoir¹⁵.

The new V isotopic data presented here, combined with existing Tl isotopic data, provide more clarity on the post-ca. 2.32 Ga oxygenation of the oceans, due to nuances in how the oxic sinks for each of these elements operates. Unlike the oxic Tl sink that is Mn oxide specific, the operation of a highly fractionated oxic V sink is dependent on the stabilization of vanadate at dissolved O_2 levels $> 10 \mu\text{M}$, which is then adsorbed onto a diffuse global flux of various Mn and (dominantly) Fe oxyhydroxides^{16,18}. This makes $\delta^{51}\text{V}_{\text{sw}}$ less sensitive to lower dissolved O_2 levels that may still have promoted major Mn oxide burial, and $\epsilon^{205}\text{Tl}_{\text{sw}}$ perturbations, in the Mn^{2+} -rich pre/syn GOE oceans⁴⁷. However, $\delta^{51}\text{V}_{\text{sw}}$ responses to ocean oxygenation above $10 \mu\text{M}$ O_2 should have been less impacted by the specifics of the marine Mn cycle in the aftermath of atmospheric oxygenation^{15,48}. Therefore, the combination of $\epsilon^{205}\text{Tl}$ records, which have more expansive seawater archives, but more specific redox drivers, and $\delta^{51}\text{V}$ records, which may respond directly to marine O_2 , but have more complex, only qualitative seawater archives and a higher oxygenation threshold, provides more texture to the history of early ocean oxygenation than either proxy can alone.”

Line 64. Explain what authigenic V is here. BTW, what you measured is V isotope composition of black shale leachate.

Clarified this point: *"In this study we targeted organic-rich shales and analyzed $\delta^{51}\text{V}$ in the authigenic V fraction ($\delta^{51}\text{V}_{\text{auth}}$). This fraction represents the V scavenged from Paleoproterozoic seawater by sinking organic matter, the isotopic composition of which allows reconstruction of relative changes in the global ocean redox state¹⁹."*

Line 68–101. The background of V isotopes as a paleo-redox proxy should be moved before "For an independent perspective on ocean oxygenation across the GOE".

We prefer the current format, where an overview of our study approach is provided before giving a (now more in depth) introduction to V isotopes as a whole.

Line 68. "open-ocean seawater" rather than "seawater"; subscript "SW" revised to "OSW". We have added this clarification to the sentence in question but choose to keep the 'sw' subscript, following common convention.

Line 80. Just Cariaco Trench, not including Black Sea. The $\delta^{51}\text{V}$ values of euxinic sediments in the Black Sea are from -0.48‰ to -0.66‰ , with an average of $-0.48 \pm 0.06\text{‰}$ (Chen et al., 2022).

Chen, X., Li, S., Newby, S.M., Lyons, T.W., Wu, F., Owens, J.D., 2022. Iron and manganese shuttle has no effect on sedimentary thallium and vanadium isotope signatures in Black Sea sediments. *Geochim. Cosmochim. Acta* 317, 218–233.

Removed reference to the Black Sea as suggested

Line 92. "0.01" or "0.1"?

0.1, thanks for catching this.

Line 95. How about hydrothermal V input? As said before, "O₂ at mid-ocean ridge depths is not expected to have been present around the GOE to support rapid hydrothermal Fe oxidation and V coprecipitation". Dong et al. (2024) has explained why the hydrothermal V input is thought to be minimal.

Dong, L.-H., Wei, W., Xu, L., Lin, Y.-B., Liu, Z.-R., Pan, S., Jing, Z., Huang, F., 2024. Vanadium isotope evidence for seawater contribution to V enrichment/mineralization in early Cambrian metalliferous black shales. *Sci. Bull.* 69, 1006–1010.

Added text: *“Hydrothermal fluid V input to the oceans is also expected to have negligible effect on the seawater isotope mass balance”* with a citation to the Dong paper.

Line 99–101. It’s better to provide some references (Fan et al., 2021; Wei et al., 2023, 2025; Heard et al., 2024) here to support this statement.

Fan, H., Ostrander, C.M., Auro, M., Wen, H., Nielsen, S.G., 2021. Vanadium isotope evidence for expansive ocean euxinia during the appearance of early Ediacara biota. *Earth Planet. Sci. Lett.* 567, 117007.

Heard, A.W., Wang, Y., Ostrander, C.M., Auro, M., Canfield, D.E., Zhang, S., Wang, H., Wang, X., Nielsen, S.G., 2023. Coupled vanadium and thallium isotope constraints on Mesoproterozoic Ocean oxygenation around 1.38–1.39 Ga. *Earth Planet. Sci. Lett.* 610, 118127.

Wei, W., Chen, X., Ling, H.-F., Wu, F., Dong, L.-H., Pan, S., Jing, Z., Huang, F., 2023. Vanadium isotope evidence for widespread marine oxygenation from the late Ediacaran to early Cambrian. *Earth Planet. Sci. Lett.* 602, 117942.

Wei, W., Wang, H., Zhang, S., Dong, L.-H., Li, D., Huang, F., 2025. Dynamic redox evolution in the middle-late Mesoproterozoic oceans. *Chemical Geology* 679, 122685.

We have added citations as suggested.

Line 105–108. Provide more details on the V isotope data of both lower and upper sections. We added slightly longer descriptions of the data, but as many of the nuances of variations, particularly in the lower section, require description when they are explained in the Discussion, it would be repetitive to list them here.

Line 108–110. These contents could be introduced earlier.

We moved this text up to the end of the paragraph introducing the EBA-2 samples that we targeted.

Line 119–121. Why not present the Fepy/FeHR data in Fig. 1?

We showed which samples were (possibly) euxinic in panels B and I using black filled circles, and find that the figure becomes unmanageably cramped when adding additional panels (we have already added a Tl isotope panel to each row).

Line 122–124. The Mo EF and U EF co-variation could help to distinguish local depositional environment.

Mo EF, U EF, and V EF all show rough positive correlations with one another in both the upper and lower sections, supporting the interpretation of an anoxic depositional environment capable of drawing down these redox sensitive elements. However, to be more specific than this using RSE EFs alone is difficult, because the seawater reservoirs of at least some of these elements were likely different than they are today. As such, using defined thresholds (such as those recommended by Bennett and Canfield, 2020) based on absolute RSE/Al or EF values would not work with their modern calibration. We rely on the Fe speciation to provide more info on euxinic vs. anoxic etc., but do see examples (two low FeHR/FeT samples near the bottom of the section) where the RSE data clarify that samples were deposited under reducing conditions despite what their Fe speciation indicates. The holistic use of all proxies available to us appears to be key to best defining the local redox conditions.

Line 125. Before reconstructing the $\delta^{51}\text{V}_{\text{SW}}$ values from sedimentary record, it is necessary to estimate whether the $\delta^{51}\text{V}_{\text{SW}}$ value is globally homogeneous, which is controlled by input flux and oceanic V reservoir. Or you just reconstruct local seawater $\delta^{51}\text{V}$ values? We think that much of this question might be addressed a few paragraphs down at the end of this section, where we discussed the open ocean connectivity of the depositional setting, and any possible directional impacts on $\delta^{51}\text{V}$ in this stratigraphy as a result of local riverine influences:

"Furthermore, despite deposition in an environment that was well-connected to the open ocean, the pro-delta setting may conceivably have allowed some degree of mixing between global seawater and UCC-like river water inputs, which could bias reconstructed $\delta^{51}\text{V}_{\text{SW}}$ towards slightly more negative values than the real seawater value³³. This bias, if present, would have more significantly impacted the samples from the top of the section that were deposited closest to shore under the shallowest paleodepths (Fig. 1), so any correction to account for this would again only increase the magnitude of the positive shift in reconstructed $\delta^{51}\text{V}_{\text{SW}}$ up-section."

Additionally, we have added some more text to the introductory sections emphasizing how V isotopes generally work as a global redox proxy because the residence time of V (90 kyr) is many tens of times longer than the ocean mixing timescale:

"Vanadium isotope geochemistry provides information on the global ocean redox state because 1) V is redox sensitive, behaving differently in different redox environments (Fig. 2); and 2) it has a long (ca. 90 kyr) residence time relative to modern and ancient ocean mixing timescales on the order of 1 kyr²⁰, such that the dissolved V reservoir and its isotopic signature

should be globally well-mixed in the open ocean and unrestricted basins¹⁹. Because of this, reconstruction of marine V isotope mass balance from sedimentary archives can shed light on the ancient ocean redox state."

This is likely to hold even if the marine V reservoir was smaller, because ocean mixing timescales in the Archean-Paleoproterozoic were similar to today and thus there are two orders of magnitude of flexibility in the V residence time before we have to start worrying about local heterogeneity.

Line 127. How about the correlation between the V EF and $\delta^{51}\text{V}$ values, which can help to estimate the effect of local V drawdown?

We replaced the previous supplementary Figure with a new one showing more relevant panels for inferring local redox (including TOC data), which includes the cross plot the reviewer refers to. While there might be a suggestion of a negative relationship between $\delta^{51}\text{V}$ and V EF in the lower section, consistent with expectations of V drawdown (e.g. Fan et al., 2021), the data are very scattered and it is hard to make the case for more precise reconstruction of seawater V isotope values. We note that all but one upper section sample has low V EF in this plot compared to the lower section, but this appears to be related to a relative lack of TOC rather than V drawdown (refer to plots with TOC on horizontal axis). Therefore, this supports our interpretation that an anoxic $\Delta^{51}\text{V} = -0.7$ is an appropriate initial correction to apply to all data, while acknowledging, as we do in the main text, that if there were local variations in $\Delta^{51}\text{V}$ throughout deposition of the section, it is more likely that the upper section would be associated with a larger magnitude $\Delta^{51}\text{V}$ correction, resulting in a larger reconstructed shift in $\delta^{51}\text{V}$, than the converse.

The new caption in the revised supplementary figure, relevant to this comment is:

"Lack of systematic covariation between $\delta^{51}\text{V}_{\text{auth}}$ and $\text{Fe}_{\text{HR}^}/\text{Fe}_T$, TOC, V EF suggests that isotopic fractionation between seawater and authigenic V was not systematically controlled by local redox conditions. A rough negative covariation of some lower section $\delta^{51}\text{V}_{\text{auth}}$ and V EF data may resemble the signatures of local seawater V drawdown seen in other anoxic Precambrian sediments⁵, but a relationship is not well developed enough to allow more precise seawater $\delta^{51}\text{V}$ reconstruction than the qualitative ranges applied in the main text. Lower V EF in the upper than lower section appears to be related to limited TOC with which to deliver authigenic vanadyl to sediment, rather than local drawdown of the seawater V reservoir."*

Line 129–132. I can see you use $\Delta^{51}\text{V}$ of -0.7‰ to reconstruct maximum $\delta^{51}\text{V}_{\text{SW}}$ value for lower section in Fig. 2. Why not discuss it here?

We believe this query is answered in the following text of the MS, we just required some more introductory/ground laying text before explaining the values we applied in each part of the stratigraphy.

Line 164–166. I think the content about the V isotope fractionation calculations (Line 479–500) in Method could be moved here.

We have made some changes to this section to make it easier to follow, per the request of Reviewer 3. However, we prefer to keep it in the Methods so as not to break the flow of the main text. The key results of this modeling are simple percentage sink abundances output from a Rayleigh distillation. As a compromise, we have added the detail that a Rayleigh distillation model was used to the main text at a relevant point.

Line 168–169. This statement is a little bit unclear as well.

Clarified with extra explanation:

“Using a simple two-component carbonate isotope mass balance, such a change in the C_{org} burial fraction could have induced a $>4\text{‰}$ drop in seawater $\delta^{13}\text{C}$ that is not present in syn-GOE carbonate records.”

Line 194–196. I suggest to present the Tl isotope data in Fig. 1.

Done. There is also now a cross plot against V isotope data later in the manuscript

Line 202. “data suggest”.

Fixed, thanks.

Line 203–204. I don’t understand this statement. “Drawdown of the shallow seawater Mn^{2+} reservoir” led to “restriction of Mn oxide burial” in the open ocean? Please clarify it.

Expanded the text here to clarify the point:

“Vanadium isotope data suggest that post-ca. 2.32 Ga, shallow marine, bottom water O_2 levels may have exceeded $10\ \mu\text{M}$ by the time of deposition of the upper section, so it is unlikely that Mn oxide burial would have been limited by O_2 availability. Therefore, the more likely interpretation of the Tl isotope record at this time is that extensive earlier Mn oxide burial had drawn down a large pre-GOE seawater Mn^{2+} reservoir to such a degree that Mn oxide burial became limited by Mn^{2+} availability¹⁵. Indeed, a possible negative correlation developed by

small $\delta^{51}V_{auth}$ and $\epsilon^{205}Tl_{auth}$ variations within the upper section may reflect the diminished 'response signal' of $\epsilon^{205}Tl_{sw}$ to small fluctuations in marine O_2 in the face of a diminished Mn^{2+} pool from which to form Mn oxides (Fig. 4)."

Line 217–219. But the oxidized V sink is actually Mn oxides. Is it possible that the adsorption efficiency of Mn oxides is different for V and Tl?

Vanadium is adsorbed onto Mn oxides, but it is also extensively drawn down in oxic environments by Fe oxyhydroxides and pelagic clays, and this 'oxic sink' with a distinct isotopic fractionation appears above a threshold of $\sim 10 \mu M O_2$. This makes it distinct from Tl, whose only fractionated oxic sink is hexagonal birnessite, a type of Mn oxide mineral. These different mineral hosts allow decoupling of V and Tl isotope systematics through changing ocean redox. The new Fig. 4 in the revised text (cross plots of V and Tl isotopes, and expected trends) helps to visualize where these systems may or may not be coupled, and we have added additional clarifying discussion on this topic in the revised text (see our response higher up for the relevant quoted text).

Figure 1. (1) I suggest to add a geologic map of the studied area at least in this figure; (2) add the FePy/FeHR data and Tl isotope data.

(1) Geological maps of the area are available in the cited references and no new mapping was performed in this study. As we present data from a single core and provide the lithostratigraphic log in Fig 1, we do not think that the map would convey any useful additional information pertinent to this particular study.

(2) We added the Tl isotope data, and explained above how FePy data are displayed, within space constraints.

Reviewer #3 (Remarks to the Author):

The manuscript, "Onset of persistent surface ocean oxygenation during the Great Oxidation Event," adds new vanadium isotope data for shales from the ~ 2.3 Ga Transvaal SGp, South Africa, which have already been extensively geochemically characterized (S isotopes, Fe speciation, Tl isotopes, whole-rock major- and trace-element geochemistry). The authors argue that over the studied interval there was a positive shift in seawater $d^{51}V$ of ~ 0.25 permil, and that this shows oxygenation of the shallow oceans to > 10 micromolar dissolved O_2 . The novelty of this conclusion is that it might demonstrate a relatively quick onset of shallow-water oxygenation following the disappearance of mass-independent fractionation of sulfur generally defined as the GOE.

We thank the reviewer for providing a clear summary of their (correct) understanding of our work, and for identifying its novelty.

This work is original and explores several interesting ideas about the oxygenation of the Earth surface environment in the early Paleoproterozoic. Overall the manuscript is clear and well written and will be of interest to a broad geological audience. I have two major concerns, which I outline below, and several minor concerns that will also need to be addressed.

Again, thank you for the positive feedback on the content and writing of the work. We hope we have addressed the outlying concerns in the revised submission to Nature Communications.

1. The major interpretation of the paper is that there was an increase in seawater $\delta^{51}\text{V}$ between the Rooihoogte Fm and Timeball Hill Fm contact and the Upper Timeball Hill Formation. In some ways, the authors were careful about this interpretation and considered a range of D^{51}V (degree of fractionation between sediment and seawater, using modern settings as analogues for the range of environments they might expect ~ 2.3 Ga), and they make the best estimate they can of the riverine value based on modern riverine value and the difference documented in Paleoproterozoic glacial till (see colored boxes in Fig. 2). My concern is that the authors were not careful about how they considered their own data in this context—they used averages without uncertainty to demarcate the differences between the lowermost section of the core and the upper section of the core. Given the large degree of variation in the lower section across the Rooihoogte and Timeball Hill Fms, this feels potentially problematic to me because if you consider the full variability in the lowermost core, the potential field for the $\delta^{51}\text{V}$ of seawater expands substantially, possibly up to -0.15 permil (0.7 permil addition to the highest $\delta^{51}\text{V}$ value measured < 1300 m). At the very least, the authors need to acknowledge this, and to consider how to show uncertainty or 2SD around the average values in Fig. 2.

Thank you for this comment – we agree, and had difficulty when preparing this figure with the decision of how to represent both sources of uncertainty (what correction to apply, and analytical uncertainty) in a single figure. We have altered this figure to account for both sources of uncertainty by adding a ‘bubble’ that shows the limits of the maximum correction factor plus 2SD analytical uncertainty for each datapoint beyond that, and overlay the average and standard deviation for the samples groups with darker vertical bars. We changed the label to ‘ 0.27 ‰ avg’ increase, to show that this shift is specifically calculated

between the two average values. There is now some overlap between the bubbles, but it is worth noting that many of the most positive $\delta^{51}\text{V}$ values in the lower section are in the euxinic samples (black circles), which might be expected to be associated with a smaller correction to recover the seawater value; and due to local redox conditions, we would expect that if there were any difference for $\Delta^{51}\text{V}$ in the upper and lower sections, there would be a larger magnitude of correction for the upper section resulting in a larger apparent shift in the seawater isotopic composition.

Similarly, the authors should offer explanation for the degree of $\delta^{51}\text{V}$ variability between 1315 and 1350 m, even if it is not important for their overall conclusion of a shift up section. While we do not see perfect correlations with any one environmental proxy, a good deal of this variability in the lower section is likely attributable to local variations in V drawdown. As shown more clearly in the revised Fig S1, a (noisy) negative relationship between V EF and $\delta^{51}\text{V}_{\text{auth}}$ is consistent with the expectations of a $\Delta^{51}\text{V}$ that decreases in magnitude where local seawater V drawdown is greater (making less V available to build up sedimentary V EF). This is similar to, though less cleanly expressed than, the observations of Fan et al. (2021) in Ediacaran shales exhibiting a range of V drawdowns that scale with $\Delta^{51}\text{V}$. Some of the $\delta^{51}\text{V}_{\text{auth}}$ values that deviate most strongly from this negative V EF vs. $\delta^{51}\text{V}_{\text{auth}}$ trend in the lower section are euxinic samples that may be expected to feature a smaller $\Delta^{51}\text{V}$, and thus generated a more positive $\delta^{51}\text{V}_{\text{auth}}$ from a given seawater composition. Last, it could be tempting to suggest that the heaviest $\delta^{51}\text{V}_{\text{auth}}$ value in the lower section, which is stratigraphically higher than the first S-MIF disappearance, could be a signal of ocean oxygenation; but the isolated nature of this datapoint, and the fact it is still well within error of the other data, makes such a speculation highly uncertain.

We have added some more elements of the above-discussed points to the main text in the revised version:

“As discussed above, the targeted samples were deposited under reducing conditions. These local conditions make it appropriate to first consider an effective $\Delta^{51}\text{V}$ between paleoseawater and sediments in a spectrum of values between -0.7 and 0.0 ‰, depending on the extent of local V drawdown (Fig. 3)¹⁹. The $\delta^{51}\text{V}_{\text{auth}}$ values show no correlation with $\text{Fe}_{\text{HR}^}/\text{Fe}_{\text{T}}$, $\text{Fe}_{\text{py}}/\text{Fe}_{\text{HR}^*}$, or total organic carbon (TOC) (Fig. S1). A possible negative co-variation with V EF occurs in the lower section, which may indicate water column V depletion (Fig. S1). However, this relationship is weak relative to that commonly seen in younger Precambrian shales³⁷, making it difficult to select a specific value for $\Delta^{51}\text{V}$ (within the range of -0.7 to 0.0 ‰) to reconstruct $\delta^{51}\text{V}_{\text{sw}}$, or to apply a variable sample-by-sample correction. Related to this, much of the scatter in $\delta^{51}\text{V}_{\text{auth}}$ in the lower section may reflect variability in the local $\Delta^{51}\text{V}$ expressed during V drawdown to sediments under reducing conditions, relative to a potentially less variable $\delta^{51}\text{V}_{\text{sw}}$ ”*

at the time of deposition. The approximate minimum reconstructed value of $\delta^{51}V_{sw}$ allowable by mass balance should be defined by the syn-GOE riverine value, as no known sinks decrease $\delta^{51}V_{sw}$ relative to inputs. Based on the ~ 0.1 ‰ lighter composition of the UCC at this time, we assume the riverine value and thus the minimum allowable paleo- $\delta^{51}V_{sw}$ was around -0.7 ‰^{33,34} (Fig. 3). Meanwhile, the maximum paleo- $\delta^{51}V_{sw}$ during deposition of the lower section is defined by a 0.7 ‰ offset to the $\delta^{51}V_{auth}$ data, giving a maximum value of -0.37 ‰ on average, with the range allowed within analytical error on individual data points extending to even higher values (Fig. 3). "

2. The samples were run in two separate analytical sessions, thus a description is needed of how the authors have accounted for the uncertainties imposed by combining data from multiple sessions (typically this requires determining and applying an excess uncertainty associated with each session, and it can possibly also include the within-lab long-term excess uncertainty, if the latter is available). Properly determined uncertainties are crucial for comparing between analytical sessions in one study and even more important for other groups who will want to make an accurate comparison between their and your vanadium isotope data in future studies.

We should have done a better job of clarifying: separate samples were run in totality in each of the two sessions. This was not a situation where the analysis of the full sample set was split over two analytical sessions and the data combined. We have clarified this point in the revised text, and we have done a better job in the revised methods section of describing the uncertainties reported for our data:

"uncertainties are presented as the larger of either the 2SD uncertainty of BDH analyses run in the same sequences or the 2SD of replicate analyses of the samples."

Beyond this, we agree that it is important to consider session-to-session variability. As can be seen from the supplementary data file, the second session (September 2023) had larger measurement errors, with the minimum error for any given sample being a 2SD of 0.15 for the BDH solution analysis. We have added a tab to the supplementary data giving the date/session in which each sample and geostandard were analyzed, and added the following text to the revised methods section:

"Our V isotope dataset was generated by analysis of two separate batches of samples over separate analytical sessions (Supplementary Data File). We saw no evidence for systematic differences in $\delta^{51}V$ values measured in each session. Of the geostandards measured in both sessions (SCo-1 leach, BHVO-2), one was lighter (but within error) and another was equal to or heavier (but within error) than the recommended value for each session, and the BDH

values were almost identical (second session slightly lighter), and always within error of the recommended value. For the EBA-2 drill-core shale samples, no upper section samples (<1300 m depth) were analyzed in the first session. However, the average value of the lower section (>1300 m) samples was identical in each session (first session: $\delta^{51}V_{auth} = -1.07 \pm 0.08$ 2SE, second session: $\delta^{51}V_{auth} = -1.07 \pm 0.11$ 2SE). Therefore the positive $\delta^{51}V$ shift (averaging 0.27‰) observed from the lower to upper section appears not to be an artifact of different analytical sessions.”

Minor comments:

-In the abstract, the rocks are referred to as 2.32-2.26 Ga, but it is unclear where the 2.26 Ga age constraint is from, and it should be referenced directly in the text.

These are U-Pb in tuff dates for the upper Timeball Hill Formation, from Rasmussen, B., Bekker, A. & Fletcher, I. R. Correlation of Paleoproterozoic glaciations based on U–Pb zircon ages for tuff beds in the Transvaal and Huronian Supergroups. *Earth Planet. Sci. Lett.* 382, 173–180 (2013).

We have now added the relevant citation at the appropriate point in the text:

“The canonical disappearance of S-MIF occurs in the upper Rooihogte Formation at ~1,340 m drill-core depth², with a depositional age of 2.316 ± 0.007 Ga⁹. Further age constraints come from U–Pb dating of two tuff beds in the nearby drill-core EBA-1, which gave ages of 2.256 ± 0.006 Ga and 2.266 ± 0.004 Ga for the upper Timeball Hill Formation, representing the top of our studied section⁴⁰.”

I cross-plotted the V and Tl isotope signatures in lower interval, and there is a lack of correlation of these signatures there. What does this mean? I think this should be addressed in the text. There are several mentions of the Tl isotope data in the manuscript, but it is not included in figures and it will help the readers to add this, if even just to the supplement.

As all reviewers commented in some way on the display or correlation of Tl isotopes along with V, we have added the Ostrander et al. Tl isotope data to the profiles in Fig. 1, and have also provided a cross plot similar to the one the reviewer here referred to, as our new Fig. 4. There is a negative correlation between Tl and V isotope data in the upper section, which we briefly discuss in the Fig. 4 caption, and in the revised text in our discussion of the coupled (or not) growth of oxic sedimentary sinks for these elements. Please see our response to Reviewer #2 detailing the extensive additions we made to the text to explain the V and Tl correlations, and where coupling between these proxies may or may not occur.

For the lower section, a possible negative relationship can also be seen if an outlier is ignored, but the lack of clear correlation can be attributed to a couple of causes:

1) As discussed in the context of growing oxic sinks near the end of the manuscript, the threshold O₂ level required to start building up a substantial oxic V sink that would change seawater V mass balance is >10 μM O₂; so it is possible that the oxic TI sink could have grown earlier (in the lower section) through the precipitation of Mn oxides at <10 μM O₂, high Mn²⁺ conditions.

2) As discussed in the section on reconstruction seawater δ⁵¹V values, there could be a variable local fractionation of V isotopes from seawater into sediment, between the values 0.0 and -0.7 for anoxic and euxinic sediments. These fractionations, which we cannot perfectly correct for on a sample-by-sample basis, are a major source of uncertainty in (now) Fig. 3, and can explain why sedimentary authigenic V isotopes do not correlate well with TI isotopes, for which shales are a more straightforward seawater archive.

-Please explain how there are two samples at ~1346 m which are apparently oxic but possibly euxinic based on Fe speciation data

Thanks for pointing this error out. We made a mistake in our plotting, plotting all datapoints with a Fe_{py}/HR* in excess of 0.6 regardless of FeHR*/FeT. These were the only two datapoints erroneously plotted as euxinic/possibly euxinic. However we actually missed blacking out one euxinic sample (with high FeHR*) around the same depth level) in some panels, and it has now been placed. We do note (here and in the text) that those two low FeHR*/FeT samples have elevated V, Mo, and U EFs and thus were likely deposited in reducing conditions despite their Fe speciation data.

-Figure 3 labels for shallow and deep oceans are confusing and imply that they are their own depth y-axis

The y axis is a qualitative measure of O₂, the labels indicate which line in the diagram corresponds to the shallow and deep oceans. We have edited this figure in a manner we hope minimizes this confusion.

Methods:

-A standard deviation is not an appropriate notation for uncertainty of a weighted average, that should be a standard error.

We used the word 'weighted' incorrectly in this context, as we used the simple 2SD for either samples or BDH solutions. We have amended the description of uncertainty to be more clear about the source of these numbers:

"uncertainties are presented as the larger of either the 2SD uncertainty of BDH analyses run in the same sequences, or the 2SD of replicate analyses of the samples."

-Reference material results (for 'geostandards') needs to be included in supplementary table.

Geostandard results and comparison to recent literature values are provided in a second tab in the revised supplement.

-"Vanadium isotope fractionation calculations" section was difficult to follow and would benefit from more sign-posting for non experts.

Thanks for this suggestion. We have added quite a lot more explanatory text to this section to help non-expert readers.

This is my second time to review this paper. Considering this is a new submission to *Nature Communications*, I go through the whole MS as a new one. Of course, this is still a very important study, and the new V isotope dataset is with high quality and scientific significance. In addition, authors have addressed most of the points raised by the previous reviewers, and the MS has been certainly improved. I appreciate their effort to take the reviewers' comments into account, and suggest publication after considering my new comments listed below.

Line 37. Could you please introduce more details on the trajectory of biological innovation during the GOE?

Line 60. I encourage authors to consider a more important scientific question, rather than complementing the Tl isotope perspective on ocean oxygenation across the GOE using V isotopes.

Line 74–77. But the residence time of oceanic V may be shorter than 90 kyr in ancient oceans.

Line 77–78. Deposition of dissolved V into sediments with different isotope fractionation should be involved in the reasons why V isotope geochemistry provides information on the global ocean redox state, which I think is the most important.

Line 86. I don't think this isotope fractionation is only controlled by Fe-oxyhydroxide.

Line 97–99. This should be indicated in Fig. 2.

Line 118. You should at least provide the location of this drill core.

Line 156. Referred to as the “upper section”?

Line 161. Again, you should make sure if the paleoseawater $\delta^{51}\text{V}$ value was globally homogeneous!

Line 165–166. I can see a robust positive correlation between the TOC and V EF, which could support the removal of oceanic V by organic matter.

Line 201–207. Maybe, some paleo-salinity proxies can help to eliminate this effect.

Line 221–226. Figures are necessary to visually illustrate the Rayleigh distillation model (Line 223) and two-component carbon isotope mass balance model, at least in Supplementary materials (Line 225–226).

Line 233–236. See the comment above.

Line 260. “These data provide”.

Line 472–488. Figs. 1H-N are not described.

Figure 1. The relationship of (A-G) with (H-N) is not shown clearly. Authors can revise as Ostrander et al. (2024).

Figure 2. The V isotope fractionation between euxinic sediments and open-ocean seawater is from -0.7‰ to 0.0‰ , controlled by local V drawdown efficiency. Please revise it.

Wei Wei

University of Science and Technology of China

Reviewer Comments

Author Response

"Quoted Text from revised MS"

Reviewer #1 (Remarks to the Author):

This is my second time to review this paper. Considering this is a new submission to Nature Communications, I go through the whole MS as a new one. Of course, this is still a very important study, and the new V isotope dataset is with high quality and scientific significance. In addition, authors have addressed most of the points raised by the previous reviewers, and the MS has been certainly improved. I appreciate their effort to take the reviewers' comments into account, and suggest publication after considering my new comments listed below.

We thank the reviewer for two rounds of useful comments and appreciate their recommendation that our manuscript be published pending these last recommended changes.

Line 37. Could you please introduce more details on the trajectory of biological innovation during the GOE?

Added as requested:

"ultimately laying the groundwork for complex multicellular life"

Line 60. I encourage authors to consider a more important scientific question, rather than complementing the TI isotope perspective on ocean oxygenation across the GOE using V isotopes. While focusing solely on the title sentence of this paragraph arguably misses the important science question we formulate within this paragraph, we take the reviewer's point and revised this first clause to read:

"However, evidence for Mn oxide burial provide only one, qualitative index for rising O₂, so to provide further texture to our understanding of ocean oxygenation across the GOE".

Line 74–77. But the residence time of oceanic V may be shorter than 90 kyr in ancient oceans.

Addressed with the added text:

"This global homogeneity is also expected to hold in the Paleoproterozoic, because V concentrations in black shales from this time period suggest a comparable order-of-magnitude size of the marine dissolved V pool to the modern^{22,23}, and thus there is no obvious reason why the oceanic V residence time would be orders of magnitude lower¹⁹. "

Line 77–78. Deposition of dissolved V into sediments with different isotope fractionation should be involved in the reasons why V isotope geochemistry provides information on the global ocean redox state, which I think is the most important.

We had added this point about isotope fractionation to point 1) of this paragraph, as suggested:

“Vanadium isotope geochemistry provides information on the global ocean redox state because 1) V is redox sensitive, behaving differently and taking on different isotopic compositions in different redox environments (Fig. 2); and 2) it has a long (ca. 90 kyr) residence time relative to modern and ancient ocean mixing timescales on the order of 1 kyr²⁰, such that the seawater dissolved V reservoir and its isotopic signature should be globally well-mixed in the open ocean and unrestricted basins¹⁹.”

Line 86. I don't think this isotope fractionation is only controlled by Fe-oxyhydroxide.

We indicated the fractionation is explainable by this mineral control, rather than that this is the sole control. However we have edited the text to be more inclusive:

“and adsorbed onto a range of Fe-Mn oxyhydroxide surfaces^{19,21}”

Line 97–99. This should be indicated in Fig. 2.

This has been fixed as requested.

Line 118. You should at least provide the location of this drill core.

This information was already in the Methods section on Geological setting of samples but sure, we have added it here too.

Line 156. Referred to as the “upper section”?

Thanks for catching this, we added the parenthetical “(hereafter referred to as the ‘upper section’)” as suggested.

Line 161. Again, you should make sure if the paleoseawater $\delta^{51}\text{V}$ value was globally homogeneous! This is now addressed in our response to requested addition above, pertaining to the residence time of V in the oceans.

Line 165–166. I can see a robust positive correlation between the TOC and V EF, which could support the removal of oceanic V by organic matter.

We agree, there is some supporting evidence for that point. We edited those sentences to read:

“A possible negative co-variation with of $\delta^{51}\text{V}_{\text{auth}}$ V EF, and apparent positive correlation of TOC with V EF, occur in the lower section, which may indicate water column V depletion that plausibly could be driven by organic matter (Fig. S1)”

Line 201–207. Maybe, some paleo-salinity proxies can help to eliminate this effect.

This is a nice suggestion for future studies to consider, but given the limited track record of paleo-salinity studies in such ancient shales, where seawater trace element ratios may have been quite different to those assumed where these techniques are applied in Phanerozoic shales, it is not something we are readily able to employ in our study at this time.

Line 221–226. Figures are necessary to visually illustrate the Rayleigh distillation model (Line 223) and

two-component carbon isotope mass balance model, at least in Supplementary materials (Line 225–226).

Line 233–236. See the comment above.

We have provided supplementary figure panels (Figure S2) to illustrate these simple fractionation calculations, as requested.

Line 260. “These data provide”.

Changed as requested

Line 472–488. Figs. 1H-N are not described.

Figure 1. The relationship of (A-G) with (H-N) is not shown clearly. Authors can revise as Ostrander et al. (2024).

Addressing both above comments at the same time: We added text to the caption:

“(H-N) show the same data as for (A-G), with the narrow stratigraphic range at the base of the section from 1,315 to 1,350 m drill core depth expanded for clarity.”

We also added a small bracket annotation to the figure indicating that the lower panels are a zoomed in inset of the upper panels, in the manner of the Ostrander et al. paper.

Figure 2. The V isotope fractionation between euxinic sediments and open-ocean seawater is from –0.7‰ to 0.0‰, controlled by local V drawdown efficiency. Please revise it.

Done

Wei Wei

University of Science and Technology of China

Reviewer #2 (Remarks to the Author):

I was reviewer #3 in the previous round of submission of this manuscript. In re-reviewing the manuscript here, I found that the responses to my queries were answered satisfactorily and that the overall clarity of the manuscript was substantially improved by the revisions suggested by all reviewers.

We thank the reviewer for this positive assessment of our revised manuscript and are pleased we were able to satisfactorily address their queries.